# Inference of Neural Dynamics Using Switching Recurrent Neural Networks

**Yongxu Zhang    Shreya Saxena**
Yale University
{yongxu.zhang, shreya.saxena}@yale.edu

## Abstract

Neural population activity often exhibits distinct dynamical features across time, which may correspond to distinct internal processes or behavior. Linear methods and variations thereof, such as Hidden Markov Model (HMM) and Switching Linear Dynamical System (SLDS), are often employed to identify discrete states with evolving neural dynamics. However, these techniques may not be able to capture the underlying nonlinear dynamics associated with neural propagation. Recurrent Neural Networks (RNNs) are commonly used to model neural dynamics thanks to their nonlinear characteristics. In our work, we develop Switching Recurrent Neural Networks (SRNN), RNNs with weights that switch across time, to reconstruct switching dynamics of neural time-series data. We apply these models to simulated data as well as cortical neural activity across mice and monkeys, which allows us to automatically detect discrete states that lead to the identification of varying neural dynamics. In a monkey reaching dataset with electrophysiology recordings, a mouse self-initiated lever pull dataset with widefield calcium recordings, and a mouse self-initiated decision making dataset with widefield calcium recording, SRNNs are able to automatically identify discrete states with distinct nonlinear neural dynamics. The inferred switches are aligned with the behavior, and the reconstructions show that the recovered neural dynamics are distinct across different stages of the behavior. We show that the neural dynamics have behaviorally-relevant switches across time and we are able to use SRNNs to successfully capture these switches and the corresponding dynamical features.

## 1 Introduction

How does complex neural activity lead to dynamic behavior? A foundational principle in theoretical neuroscience suggests that the computations within the nervous system can be explained through the dynamics of the underlying non-linear systems (Breakspear [2017], Pandarinath et al. [2018], Durstewitz et al. [2023]). As our ability to record from larger areas of the brain with unprecedented spatial and temporal resolution increases, we find neural population activity often exhibits distinct dynamical features across time. These features may correspond to distinct internal processes or behavior (Saxena and Cunningham [2019], Churchland et al. [2012]). Identifying these dynamical features may help us understand how cognitive functions are implemented in the brain. Previous works have revealed a number of computational strategies through analysis of neural dynamics, particularly across tasks, and during orchestration of precise behavioral states such as during movement generation (Kaufman et al. [2014], Miri et al. [2017]).

State space modeling is a promising analytical approach for characterizing dynamics of time series; such models can be highly effective and have received substantial attention over many decades on analyzing temporal neural dynamics (He et al. [2023], Durstewitz et al. [2023], Sani et al. [2021]). Well-established methods on inference and learning algorithms have contributed towards learning the parameters of state space models (Blei et al. [2017], Salimans et al. [2015], Kingma and

Welling [2013], Linderman et al. [2017], Fox et al. [2008]). Neural dynamics can be time-varying due to internal fluctuations of physiological states as well as the external effect of environment. Previous works such as (Kaufman et al. [2014], Miri et al. [2017]) have found differences in neural subspaces between distinct behavioral states. Furthermore, correlation-based analysis methods, such as functional connectivity, also reveal that the brain-wide covariation in neural activity changes during movement. For example, West et al. [2022] highlight the changes in functional connectivity in the cerebral cortex, representing a series of changes in the cortical state from rest to locomotion and on return to rest. In practice, researchers typically model switching temporal dynamics by fitting different model parameters to consecutive temporal windows of neural activity (He et al. [2023], Mitelut et al. [2022], Song et al. [2022]).

In the past decade, we have seen that neural dynamics can be broadly considered to be non-linear and in a lower dimensional subspace of the recorded neural activity (McKenna et al. [1994], Rigotti et al. [2013], Hernandez et al. [2018], Saxena and Cunningham [2019], Cunningham and Yu [2014]). Accordingly, Recurrent Neural Networks (RNNs) have shown efficiency in modeling dynamics due to their non-linearity (Mante et al. [2013]). While goal-driven models have been very useful to build in normative function in these networks, data-driven RNNs are becoming more popular over recent years (Durstewitz et al. [2023], Perich et al. [2020], Duncker and Sahani [2021], Valente et al. [2022]). Specifically, these RNNs can be trained to reproduce temporal neural activity from large-scale neural recordings across a set of trials. They are also able to capture a very high amount of explained variance of the neural activity because of their static non-linearities (Durstewitz et al. [2023], Perich and Rajan [2020]). Once trained, the internal mechanisms of the RNNs can be analyzed, thus extracting the structure of the neural dynamics. Prior works have studied RNNs that perform a wide range of tasks, e.g., cognitive, sensory, motor tasks, and so on (Dubreuil et al. [2022], Mastrogiuseppe and Ostojic [2018]). However, with a few notable exceptions, models do not typically express neural activity as a set of switching nonlinear functions and directly reproduce recorded neural data. Here, we develop Switching Recurrent Neural Networks (SRNNs) which, through switches between different time-varying weights as determined by Markov transitions, are able to directly model neural activity as emanating from a discretely changing set of low-dimensional dynamical models. We perform end-to-end training of SRNNs to reconstruct the neural observations through Variational Inference. We validate SRNNs on a simulated dataset, and then analyze the performance of SRNNs on three different experimental data with distinct recording modalities and behavioral tasks with different animals: (1) electrophysiological recordings of single-unit MC activity from a non-human primate performing a reaching task (Churchland et al. [2012]), (2) cortex-wide widefield calcium imaging (WFCI) from mice performing a complex self-initiated decision-making task (Musall et al. [2019]), and (3) WFCI from mice performing a simple self-initiated lever-pull task (Mitelut et al. [2022]). We are able to not only reconstruct the multidimensional neural activity but accurately predict forward in time and recover behaviorally-relevant switches of neural dynamics. Finally, we visualize the neural dynamics in flow fields plotting and find they are distinct among different states.

## 2 Related Work

**Data-Driven Neural Networks** An artificial neural network trained to directly reconstruct observed neural activity is called a data-driven neural network. Prior studies have concentrated on employing various models to extract the underlying neural dynamics and characterize the connectivity between different components of the brain (Perich and Rajan [2020], Mastrogiuseppe and Ostojic [2018]). In Perich and Rajan [2020], the authors introduce Current-Based Decomposition (CURBD), an approach for inferring brain-wide interactions using data driven RNNs that directly reproduce experimentally-obtained neural data. In Mastrogiuseppe and Ostojic [2018], the authors used low-rank RNNs trained on high-dimensional neural activity obtained from different tasks and successfully characterized the neural dynamics and connectivity between neurons. Importantly, previous work does not consider switches in the underlying low-dimensional neural dynamics; in our work, we focus on the extraction of low-dimensional neural dynamics based on *switching* data-driven neural networks.

**Switching State Space Models** Hidden Markov Model (HMM) is a commonly-used statistical model based on a Markov process: it dictates the evolution of observations where the evolution is determined by internal factors, which are not directly observable. HMMs are widely used in computational neuroscience to identify different neural events (Baldassano et al. [2017], Masaracchia et al. [2023], Baldassano et al. [2016]). Previous studies have also explored switching dynamical systems, with the observations emanating from a dynamical system that discretely switches according

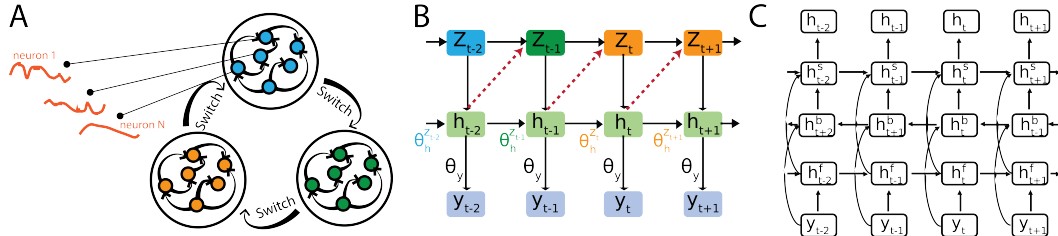

Figure 1: **Switching Recurrent Neural Network (SRNN)** (A) Schematic showing an SRNN with neural outputs. (B) Structure of the generative model for SRNN. (C) Structure of Inference Neural Network for SRNN. The $h_t^f$ and $h_t^b$ represent the states of a bidirectional RNN, followed by a $h_t^s$, which model the inferred $h_t$ from the given observation $y_t$.

to a Markov process. Auto-regressive HMMs and Switching Linear Dynamical System (SLDS) are commonly used models, where the discrete state transitions and observation dynamics are both linear and dependent solely on the previous timepoints. Various extensions of these also applied changes to these models. For example, in Linderman et al. [2017], the authors make the current discrete state dependent on the previous continuous state, termed recurrent-SLDS (rSLDS). Researchers have also changed the dynamics process to be nonlinear, such as in a Structured Variational Autoencoder (SVAE) (Johnson et al. [2016]). Moreover, both nonlinear transitions and nonlinear dynamics processes were also proposed, such as in a Switching Nonlinear Dynamical System (SNLDS) (Dong et al. [2020]). Out of these, SRNNs have the most comparable structure to SNLDS; the main differences are: (1) we specify the dynamical system as an RNN instead of passing the data to neural networks with nonlinear activation functions; (2) our transition process is the same as rSLDS, which is dependent on the previous hidden state instead of the observation in SNLDS; (3) we constrain the structure of the models in order to predict future observations. Moreover, here we analyze the ability of the model to reconstruct and elucidate the structure of recorded neural activity, while examining prediction as well as region-to-region interactions (Appendix). A relevant recent study also explored the communication between brain regions via multi-regional switching dynamical systems (MR-SDS) (Karniol-Tambour et al. [2023]), which reveals the relationship between brain regions but does not focus on analyzing behaviorally-relevant neural dynamical states and characterizing distinct neural dynamics.

**Variational Inference** Researchers have proposed numerous powerful algorithms for optimizing the parameters of Switching State Space Models. A well-known approach is the Expectation-Maxmization (EM) algorithm, which is to find maximum likelihood estimates of parameters in State Space Models, e.g., Baum-Welch Algorithm for HMMs (Bishop and Nasrabadi [2006]). Another common approach to estimating the parameters of Switching State Space Model is Variational Inference (VI), which turns the problem of computing conditional distributions of latent variables into an optimization problem that can be efficiently solved by gradient descent. Essentially, this consists of maximizing the Evidence Lower Bound (ELBO) to learn the parameters (Kingma and Welling [2013], Linderman et al. [2017]). The reparameterization trick is a crucial technique in VI, which enables more efficient gradient estimation during optimization. Instead of sampling from the variational distribution itself, reparameterization models the parameters of a distribution, usually Gaussian. This allows the gradient of the objective function with respect to the variational distribution parameters to be computed directly, and backpropagation to be implemented easily. To enable the model to generalize effectively to unseen data, we use Amortized VI. Instead of optimizing for single trials of neural activity independently as in traditional VI, Amortized VI allows us to train the models on multiple trials simultaneously, thus creating a more stable mapping between the observations and the latent variables (Dong et al. [2020], Ganguly et al. [2023], Margossian and Blei [2023]).

## 3 Methods

### 3.1 Switching Recurrent Neural Network (SRNN)

A common switching state space model is:

$$p_\theta(y, h, z) = p_{\theta_h}(h_1|z_1)p_{\theta_y}(y_1|h_1)\prod_{t=2}^{T}p_{\theta_z}(z_t|z_{t-1})p_{\theta_h}(h_t|h_{t-1}, z_t)p_{\theta_y}(y_t|h_t) \tag{1}$$

where $z_t \in \{1, ..., K\}$ is the discrete hidden states or labels, controlling the switch of the dynamics, $h_t$ is the continuous latent state with dimensionality $P$, and $y_t$ is the observation or output; in this work, the observations are directly the neural activity. We build a data-driven model with neural time-series data $y_{1:T} \in \mathbb{R}^{N \times T \times R}$ from $N$ sequences, $T$ time points and $R$ different neural dimensions as the outputs. Here, we implement a transition network, a set of dynamical networks with recurrent layers and a nonlinear activation function, and finally a linear emissions network to reconstruct the observed neural activity. We show the structure of SRNN in Figure 1A; following are the equations of the generative network:

$$p_{\theta_z}(z_t|z_{t-1}, h_{t-1}) = Cat(Softmax(f(\theta_z h_{t-1}))) \tag{2}$$

$$p_{\theta_h}(h_t|h_{t-1}, z_t = k) = \mathcal{N}(h_t|f(\theta_{h_k} h_{t-1}), \sigma_h) \tag{3}$$

$$p_{\theta_y}(y_t|h_t) = \mathcal{N}(y_t|\theta_y h_t, \sigma_y) \tag{4}$$

where $Cat$ represents Categorical distribution, $Softmax$ represents Softmax activation function, $\mathcal{N}$ represents Gaussian distribution, and $\sigma_h$ and $\sigma_y$ are the covariance matrices of the relevant Gaussian distributions. $\theta_z$, $\theta_h$, and $\theta_y$ are parameters of transition networks, dynamical networks, and emission networks respectively, which are learned through gradient descent. Here, we consider the nonlinearity $f(\cdot) = \tanh(\cdot)$ as commonly done in neuroscience in order to model the saturation effect of firing rates Pandarinath et al. [2018], Saxena et al. [2022].

## 3.2 Inference of SRNNs

We use VI to learn the parameters of our model. Here, we maximize the posterior:

$$q_{\theta, \phi}(z_{1:T}, h_{1:T}|y_{1:T}) = q_\phi(h_{1:T}|y_{1:T}) p_\theta(z_{1:T}|h_{1:T}, y_{1:T}) \tag{5}$$

where $q_\phi(h|y)$ is the posterior of continuous hidden states given the observation, $\phi$ is the parameters of our inference network. $p_\theta(z|h, y)$ is the Bayesian posterior of discrete hidden states given the continuous hidden states and the observations, which we compute using the generative network via Bayes' rule.

**Inference Network and $q_\phi(h_{1:T}|y_{1:T})$** We show the structure of our inference network in Figure 1B; we build an inference network to model $q_\phi(h|y)$ as the following:

$$q_\phi(h_{1:T}|y_{1:T}) = \prod_{t=1}^{T} q(h_t|y_{1:T}) := \prod_{t=1}^{T} \mathcal{N}(h_t|\mu_t, \sigma_t) \tag{6}$$

where $\mu$ and $\sigma$ are the mean and variance of a Gaussian distribution. Here, we use the reparameterization trick to infer the continuous hidden states Kingma and Welling [2013]:

$$h_t = \mu_t + \epsilon \sigma_t \tag{7}$$

where $\epsilon \in \mathcal{N}(0, 1)$ is a Gaussian noise parameter. The mean $\mu$ and variance $\sigma$ are typically learned via training neural networks Kingma and Welling [2013]. However, here, we do not find apparent improvement with learning the $\sigma$ in our work. Therefore, as done in previous studies, we set the variance $\sigma$ to be constant to reduce the complexity of the model and only optimize the $\mu$ Dong et al. [2020], Ganguly and Earp [2021]; we use $0.0001$ for $\sigma$. Since our generative model is based on RNNs, we make $h_t$ depend on $h_{t-1}$ to match the recurrence property. Thus, we have

$$\mu_t = f(y_{1:T}, h_{t-1}) \tag{8}$$

We pass $y_{1:T}$ to a bidirectional recurrent neural network followed by forwarding the output to a standard recurrent neural network to model Equation 8.

**Evidence Lower Bound and $p_\theta(z_{1:T}|h_{1:T}, y_{1:T})$** Typically, to learn the parameters via VI, we maximize the evidence lower bound (ELBO). Here, the ELBO is defined as:

$$ELBO = \mathbb{E}_{q_{\theta, \phi}(z_{1:T}, h_{1:T}|y_{1:T})}[\log p_\theta(y_{1:T}, h_{1:T}, z_{1:T}) - \log q_{\theta, \phi}(z_{1:T}, h_{1:T}|y_{1:T})] \tag{9}$$

With Equation 1 and Bayes' rule, we get

$$ELBO = \mathbb{E}_{q_\phi(h_{1:T}|y_{1:T}) p_\theta(z_{1:T}|h_{1:T}, y_{1:T})}[\log p_\theta(y_{1:T}, h_{1:T}) p_\theta(z_{1:T}|y_{1:T}, h_{1:T})$$
$$- \log q_\phi(h_{1:T}|y_{1:T}) p_\theta(z_{1:T}|h_{1:T}, y_{1:T})] \tag{10}$$
$$\approx \mathbb{E}_{q_\phi(h_{1:T}|y_{1:T})}[\log p_\theta(y_{1:T}, h_{1:T}) - \log q_\phi(h_{1:T}|y_{1:T})] \tag{11}$$

where the discrete hidden states $z_{1:T}$ have been marginalized out given $\sum_{k=1}^{K} p(z_t = k) = 1$ (Murphy and Russell [2001], Dong et al. [2020]).

**Initialization** Previous studies have found that a generative model trained by VI may become stuck in a single discrete state (Alemi et al. [2018], Dong et al. [2020]). Researchers have explored various methods to solve this problem, such as initialization of the transitions parameters using the parameters of auto-regressive HMM in Linderman et al. [2017] and entropy regularization in Dong et al. [2020]. In our work, we address this by initializing our model using states from an HMM trained on the same training data, which encourages the model to utilize all states and optimize the weights corresponding to each of the states. Practically, during a first phase of training, we add a term to the loss that penalizes the posterior if it does not utilize all the discrete hidden states of the HMM. In the second phase of training, we remove this penalization and train the model with our loss as described in Equation 15.

**Optimization** We use gradient descent to optimize the parameters $\theta$ and $\phi$. Given the fact that we use reparameterization to model $q_\phi(h_{1:T}|y_{1:T})$, and $q_\phi(h_{1:T}|y_{1:T})$ is independent to $\theta$, the gradient can be approximated as following:

$$\nabla_{\theta,\phi} ELBO \approx \nabla_{\theta,\phi} \log p_\theta(y_{1:T}, h_{1:T}(\phi)) - \nabla_\phi \log q_\phi(h_{1:T}|y_{1:T}) \tag{12}$$

where $h_{1:T}(\phi)$ is the output of inference network, therefore, is dependent on $\phi$. Since $\sum_z p(z){=}1$, therefore, $\log p_\theta(y_{1:T}, h_{1:T}(\phi)) = \sum_z p(z) \log p_\theta(y, h|z) = \mathbb{E}_{p(z)} \log p_\theta(y, h|z)$, and with Bayes' rule $p_\theta(y, h|z) = \frac{p_\theta(y,h,z)}{p(z)}$, Equation 12 is equivalent to:

$$\nabla_{\theta,\phi} ELBO \approx \mathbb{E}_{p(z)} \nabla_{\theta,\phi} \log p_\theta(y_{1:T}, h_{1:T}, z_{1:T}) - \nabla_\phi \log q_\phi(h_{1:T}|y_{1:T}) \tag{13}$$

In Equation 13, the first term can be rewritten by the Markov Chain.

$$\mathbb{E}_{p(z_{1:T})} \nabla_{\theta,\phi} \log p_\theta(y_{1:T}, h_{1:T}, z_{1:T}) = \sum_{k=1}^{K} p(z_1 = k) \nabla \log p(z_1 = k) p(h_1|z_1 = k) p(y_1|h_1)$$

$$+ \sum_{t=2}^{T} \sum_{k=1}^{K} \sum_{j=1}^{J} p(z_t = k, z_{t-1} = j) \nabla \log p(z_t = k|z_{t-1} = j) p(h_t|h_{t-1}, z_t = k) p(y_t|h_t) \tag{14}$$

where we use the posterior probability of the discrete hidden states, e.g., $p(z_1 = k|h_{1:T}, y_{1:T})$ and $p(z_t = k, z_{t-1} = j|h_{1:T}, y_{1:T})$, which can be obtained easily through Baum-Welch algorithm. The transition, dynamics, and emission are defined in Equation 2, Equation 3, and Equation 4 respectively. The second term of Equation 13 can be optimized via backpropagation through time (BPTT) of the inference network. In summary, with Equation 11, Equation 13, and Equation 14, our loss function is:

$$\ell = - \sum_{k=1}^{K} p(z_1 = k) \log p(z_1 = k) p(h_1|z_1 = k) p(y_1|h_1)$$

$$- \sum_{t=2}^{T} \sum_{k=1}^{K} \sum_{j=1}^{J} p(z_t = k, z_{t-1} = j) \log p(z_t = k|z_{t-1} = j) p(h_t|h_{t-1}, z_t = k) p(y_t|h_t)$$

$$- H(q_\phi(h_{1:T}|y_{1:T})) \tag{15}$$

where we use the fact that $-\mathbb{E}_{q_\phi(h_{1:T}|y_{1:T})} \log q_\phi(h_{1:T}|y_{1:T})$ is the entropy $H$ of $q_\phi(h_{1:T}|y_{1:T})$. We minimize the loss function via Adam optimizer. Our implementation is based on Pytorch 2.2.1 and we train our models using NVIDIA A100 GPUs. We provide the analysis and the results in this paper; the original code for the entire SRNNs framework on Pytorch has been made public `https://github.com/saxenalab-neuro/SRNN`.

### 3.3 Analysis of SRNNs

**Dynamical Features**: Fixed points and other dynamical features are crucial for understanding how nonlinear dynamical systems process sequences and maintain information (Sussillo and Barak [2013], Golub and Sussillo [2018]). We visualize the flow fields of RNNs, which provides a graphical representation of the evolution of network states over time. To compute these flow fields, we train our model on the entire dataset (without separating a training set from a test set) in order to capture neural dynamics across the entire data. We use $10,000$ initial hidden states uniformly sampled in a grid to compute the flow fields.

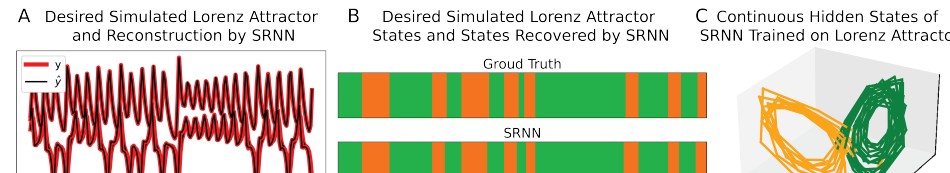

Figure 2: **Lorenz Attractor**: (A) Reconstruction using an SRNN on simulated data (thick red is the original data, thin black is the reconstruction by SRNN. The reconstruction is almost overlapped with the ground truth). (B) Comparison between ground truth discrete states and SRNN-recovered discrete states. (C) Plot of latent dynamics in each discrete state shows recovery of dynamics.

**Neural Activity Prediction**: Given the fact that SRNNs are generative models, we test all models on prediction of future neural activity using past observations. Specifically, the models are provided a time-series of previously unseen neural activity $y_{1:t}$, with which we can infer discrete hidden states $z_{1:t}$ and continuous hidden states $h_{1:t}$. Next, we sample $z_{t:t+K}$ and $h_{t:t+K}$ through transitions of discrete states and dynamics of continuous states, i.e., using Equations 2 and 3, respectively. Finally, we predict the neural activity in the future time points $y_{t:t+K}$ using the emissions network, i.e., Equation 4. Here, we change the bidirectional layers of our inference network to standard forward recurrent layers to match the prediction tasks (see Figure 1C). We explore the prediction performance by giving the model an input in different lengths ($t \in [t_0, T]$, here we use $t_0 = 10$ timepoints), and we also explore the prediction performance by predicting different lengths of neural activity ($K \in [10, 20, 30, 40]$ timepoints).

## 4 Results

We detail the results of SRNNs on inferred neural dynamics of a simulated dataset and three diverse experimental datasets ranging from monkey electrophysiology to mouse calcium imaging. On each dataset, we do $N$-fold cross-validation, where $N$ equals to the number of conditions, sessions, or subjects in the dataset. All the results in this section are reported on the test set. Additionally, we compare our method with other switching dynamical systems: Switching Linear Dynamical Systems (SLDS), recurrent Switching Linear Dynamical Systems (rSLDS) (Linderman et al. [2017]), Switching Non-linear Dynamical Systems (SNLDS) (Dong et al. [2020]), and multi-regional Switching Dynamical Systems (mrSDS) (Karniol-Tambour et al. [2023]). Additionally, we provide comparisons with Latent Factor Analysis with Dynamical Systems (LFADS) in Appendix Figure D.6. Lastly, we show example curves of the training loss, reconstruction MSE on validation data, and discrete states recovery error on validation data for SRNNs across different epochs for all three experimental datasets in Figure D.7.

### 4.1 Simulated data: Lorenz Attractor

We first apply SRNN on a well-known simulated dynamical system, the Lorenz attractor, which is famous for its "butterfly" shape (see Appendix A for relevant equations; the dimensionality of the data is $R = 3$). The dynamics can be expressed as a 2-state switching system. In our work, we generate the Lorenz data using the method and code in (Linderman et al. [2017], Dong et al. [2020]). In Figure 2, we show that SRNNs are able to reconstruct the Lorenz attractor precisely, and recover the switching states correctly, as well as identify the underlying dynamics in each discrete state.

### 4.2 Experimental data: Electrophysiology Recordings during a Monkey Reaching Task

We explore switching models on the firing rates of single units recorded from monkey motor and premotor cortex ($R = 180$), while the monkey performs a reaching task in 27 reaching conditions given targets at different locations. More details can be found in Appendix B and Churchland et al. [2012]. We have 5 different behaviorally-relevant states: (1) resting; (2) delay before the 'go' cue; (3) reaction time (from receiving 'go' cue to the beginning of movement); (4) movement execution; (5) hold at the target. Using SRNNs, we recover these states purely using the neural activity.

Firstly, we train SRNNs and competing methods on the two different types of reaching conditions (curved reaches and direct reaches) separately. We use $N$-fold cross-validation, where $N$ equals to the number of conditions in the data; $N = 18$ for curved reaches and $N = 9$ for direct reaches. We show an example of comparison between neural dynamical states recovered by different models and the behavioral states, in Figure 3A and Figure D.1A. The neural dynamical states recovered

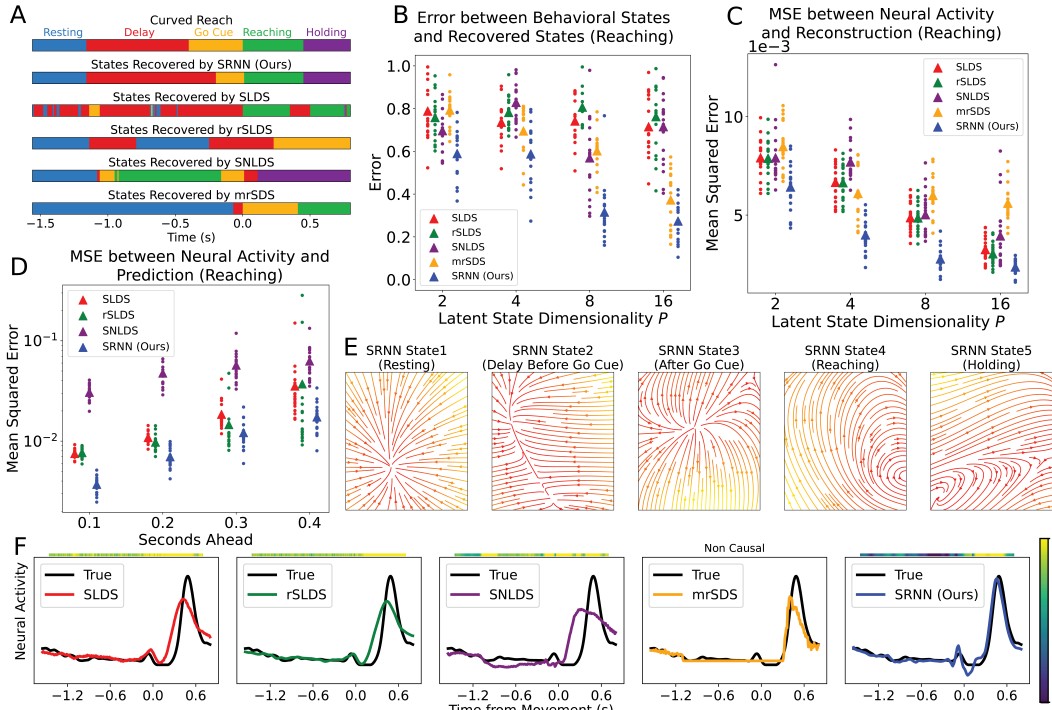

Figure 3: **Curved Reaching**: (A) Examples of comparison between behavioral states and neural dynamical states recovered by SRNNs, SLDS, rSLDS, SNLDS, and mrSDS. (B) Error between behaviorally-relevant states and recovered neural dynamical states; each dot represents one session, with the triangle as the mean of all sessions. (C) Mean squared error (MSE) between neural activity and the reconstruction. (D) MSE between neural activity and the *prediction* forward in time. (E) Flow fields of latent dynamics of SRNNs in different discrete states. (F) Reconstruction of neural activity and the corresponding ground truth for one example neuron for three models, the top color bars represent the performance on $0.1$ seconds ahead prediction. More examples are shown in Figure D.2. All results for direct reaching are shown in Figure D.1.

by SRNNs in both curved and direct reaching match their corresponding behavioral states. We use $P = 16$ for the latent state ($h$) dimensionality for all three models in Figure 3A. We quantify the difference between the recovered neural dynamical states and the behavioral states by considering them an imbalanced multi-class classification, i.e., one state as one class, the error of the classification measures the difference between the neural and behavioral states. Specifically, it is defined as the average of '1-recall' obtained on each class (Mosley [2013]). We claim that $0$ error between the recovered neural dynamical states and behavioral states does not necessarily mean that the recovered neural dynamical states are perfectly identified, because the neural dynamical switches may happen consistently leading to or following the behavioral state switches. However, these states recovered by the models with small errors can still be considered as behaviorally-relevant states, which can help us understand the cognitive function implemented in the brain. We show the error for different hidden states in Figure 3B, which illustrates that SRNNs outperform SLDS, rSLDS, SNLDS, and mrSDS on recovering the behavior-relevant neural dynamical states for $4$ different values for the latent state dimensionality $P$. We determine the number of discrete latent states via a hyperparameter sweep for SRNNs. We consider the following metrics to compare the performance of SRNNs with different $K$: (1) **Convergence to lower number of discrete states and reuse of discrete states**: We tested our model by changing the number of hidden states $K$ while keeping the number of continuous latent states $P$ constant. We show a comparison in Figure D.3A. We found that 61% of SRNNs with a higher number of discrete hidden states (e.g., $K = 6$) finally converge to the optimal number of discrete hidden states, i.e., $K = 5$, and 94% of SRNNs with a lower number of discrete hidden states (e.g., $K = 4$) had at least one hidden state reused after other states; in other words, SRNNs are not able to perform well with 4 unique discrete hidden states without reusing one of them. (2) **Reconstruction performance plateau**: While keeping other hyperparameters constant, the reconstruction accuracy plateaus at the same number of discrete states as in the behavior. Thus, we can set $K$ as the minimum number of discrete states as it takes for the model to perform well. We

show that SRNNs with $K = 5$ have lower reconstruction error in Figure D.3B. Moreover, we also implemented a 'co-smoothing' analysis (Yu et al. [2008]Karniol-Tambour et al. [2023]); we show the results in Figure D.3C, where we found that $K = 5$ also does well in reconstructing the data with a 'co-smoothing' neuron drop-out analysis. (3) **Variability across conditions**: In stereotyped tasks or experiments, such as reaching, there may not be a significant amount of variability in the timing of behavior across conditions, and this variability can thus be used as a metric for determining the optimal number of discrete states. Here, we found that SRNNs with $K = 5$ have much lower variability on recovered behaviorally-relevant states than $K = 4$ and $K = 6$ (i.e., 0.098 for $K = 5$, 0.384 for $K = 4$, and 0.282 for $K = 6$ in Figure D.3D).

We show the performance of all models on the reconstruction of the neural activity in Figure 3C; the mean squared error (MSE) between desired neural activity and the reconstruction by models indicate that SRNNs also outperform other models on reconstruction of neural activity. We note that as $P$ increases, the MSE difference between SRNNs and other models becomes smaller. Next, we show the prediction capability of future neural activity by SRNNs and competing methods in Figure 3D. We found that the prediction capabilities of SRNNs outperform competing models with linear dynamics. In addition, we show the comparison of behavioral states recovering and neural activity reconstructing performance between SRNNs with bidirectional recurrent inference networks and SRNNs with forward recurrent networks in Figure D.8. Moreover, we show the reconstruction of neural activity along with one example neuron with the corresponding ground truth in Figure 3F, with more example neurons shown in Figure D.2A-B. Here, the color bars represent performance of neural prediction. Furthermore, as a non-switching comparison model, we train an LFADS network (Pandarinath et al. [2018]) on the reaching datasets. We show reconstruction MSEs in Figure D.6A, where we use $P = 16$ for SRNN, SLDS, and rSLDS, and $P = 64$ for LFADS to keep a similar number of overall parameters in the hidden layers. We found that all three switching models have lower reconstruction error than LFADS (Figure D.6B).

We visualize the neural dynamics by plotting the flow field of the latent states $h$ in each discrete SRNN state in Figure 3E, where we used $P = 16$ based on Figures 3B,C. In order to visualize the flow fields in 2D, we apply principal component analysis (PCA) on the latent states of SRNNs to decrease the dimensionality of the flow fields to 2D. We see that the neural dynamics in different discrete states are distinct in Figure 3E: the plots reveal that the neural dynamics change from relatively stable in the first two states, to unstable after the 'go cue', and finally oscillatory during the execution and end of movement. We also show the flow fields for direct reaches in Figure D.1, and we found that the neural dynamics between curved and direct reaches are comparable during the first three discrete states, but their dynamics become different in 'movement' states, which matches the difference between the reaching type.

### 4.3 Experimental data: Widefield Calcium Imaging during Mouse Decision-Making

We show the performance of SRNNs on a decision-making task, with neural activity recorded using widefield calcium imaging (WFCI). The number of neural components is $R = 210$. The details of this dataset and preprocessing can be found in Appendix B of this paper and Musall et al. [2019]. In this data, mice have 5 behaviorally-relevant states, i.e., (1) from baseline to holding of the handle; (2) holding the handle; (3) stimulus presentation; (4) delay from the end of stimulus to the start of licking; (5) licking the spout.

Here, we use $N = 8$ pseudo-sessions, and perform $N$-fold cross-validation. In Figure 4A, we show an example of comparison between neural dynamical states recovered by different models, along with the behaviorally-relevant states described above. Here, we use a latent dimensionality of $P = 8$ for all models. We also quantify the discrete state recovery using the same approach as described in the previous section in Figure 4B, where we found that SRNNs are able to identify 5 behaviorally-relevant neural dynamical states, especially as the latent state dimensionality increases. Here, we use the same approach as in reaching task to determine $K$. We show the performance of SRNNs with different $K$ in Figure D.4, we also found that SRNNs with $K = 6$ converge to lower number of discrete states. Additionally, SRNNs with $K = 5$ perform better in the co-smoothing test in Figure D.4C. Indeed, we found that SRNNs with $K = 4$ have better reconstruction performance. However, in Figure D.4D, we found that SRNNs with $K = 5$ have much lower variability on recovered behaviorally-relevant states than $K = 4$ and $K = 6$. Therefore, given the visualization of recovered states, co-smoothing test, and variance of recovered states across different pseudo-sessions, we consider $K = 5$ as the best hyperparameter. Furthermore, we show the reconstruction performance of all models in Figure

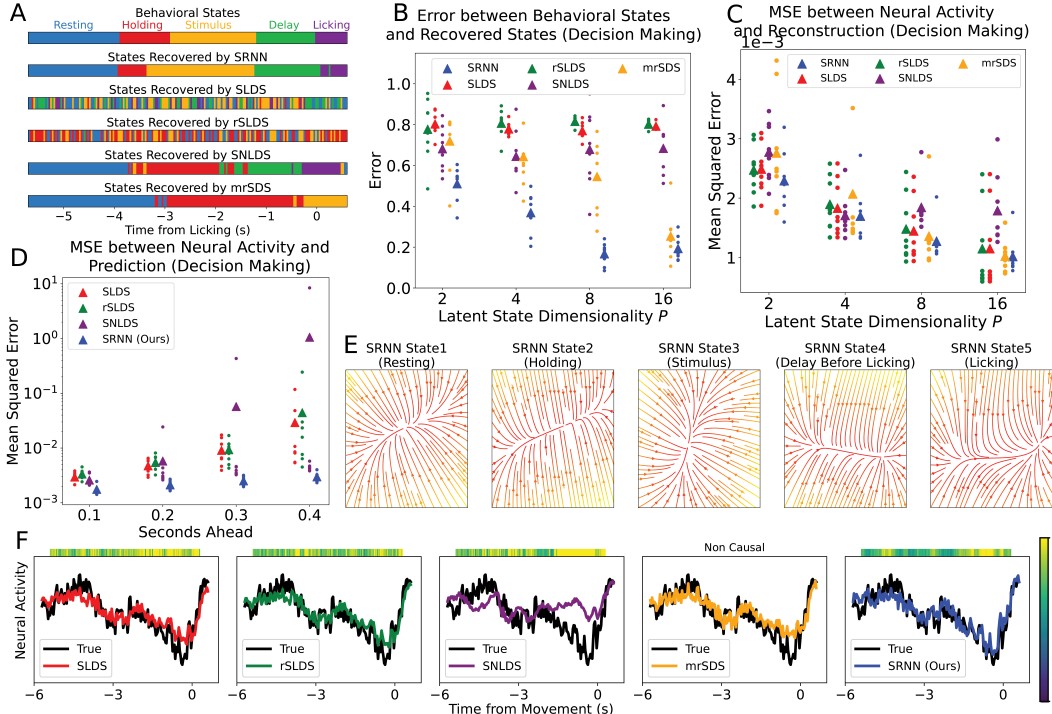

Figure 4: **WFCI Decision-Making**: (A-F) Legend similar to Figure 3 legend. The top color bars in (F) represent $0.33$s ahead prediction accuracy; more examples shown in Figure D.2.

4C, with the SRNN slightly outperforming other models. Analogous to the reaching datasets, a higher latent dimensionality results in better reconstruction. Notably, the prediction capabilities of SRNNs far outperform competing models with linear dynamics (Figure 4D). In addition, we show the reconstruction of neural activity for an example component with the corresponding ground truth in Figure 4F, with more results shown in Figure D.2C in the Appendix. Furthermore, we visualize the neural dynamics in Figure 4E with $P = 8$. We again apply PCA to visualize the flow fields, where we see that the flow fields of different RNNs are also distinct. However, the difference is smaller as compared with the reaching dataset, which may reveal that WFCI may have fewer change in dynamics over the course of a trial, potentially due to the slow timescales of calcium indicators.

### 4.4 Experimental data: Widefield Calcium Imaging during Mouse Self-initiated Lever Pull

Finally, we explore the utility of SRNNs on another WFCI data, where large-scale neural activity from the mouse dorsal cortex is recorded while mice are engaged in a spontaneous lever-pull behavior for water reward ($R = 16$). More details can be found in Appendix B and Mitelut et al. [2022]. Here, while the observed behavior happens in a very short time because the mice pull the lever very fast, Mitelut et al. [2022] found that the neural activity has obvious inhibitory dynamics starting from around 5 seconds before the self-initiated lever pull behavior. Thus, the mice have 3 behaviorally-relevant states: (1) from resting to the starting of inhibitory dynamics (as found in Mitelut et al. [2022]); (2) from the starting of inhibitory dynamics to behavior onset; (3) lever-pull execution.

We use data from 6 different mice, and perform 6-fold cross-validation. In Figure 5, we show the results with the same quantification methods as in the datasets above. SRNNs also outperform in recovering the behaviorally-relevant neural dynamical states; an example is shown in Figure 5A. While SRNNs can recover the inhibitory dynamics of neural activity from around 3 seconds before the behavior onset, SLDS and rSLDS can only identify a short state before behavior onset. We use $P = 2$ hidden states for this example since more hidden states do not improve the recovery of these discrete states (Figure 5B). We use the same approach as above to determine $K$. We show the performance of SRNNs with different $K$ in Figure D.5, where we found that SRNNs with $K = 2$ are not able to detect the switches before the behavior onset. Additionally, SRNNs with $K = 3$ have lower reconstruction error in Figure D.5B and perform better in the co-smoothing test in Figure D.5C. Moreover, we found that SRNNs with $K = 2$ have relatively lower variability on recovered behaviorally-relevant states than $K = 3$ and $K = 4$ in Figure D.5. However, as demonstrated in the state visualizations in Figure D.5A, this is primarily due to the majority of time points being classified as a single state.

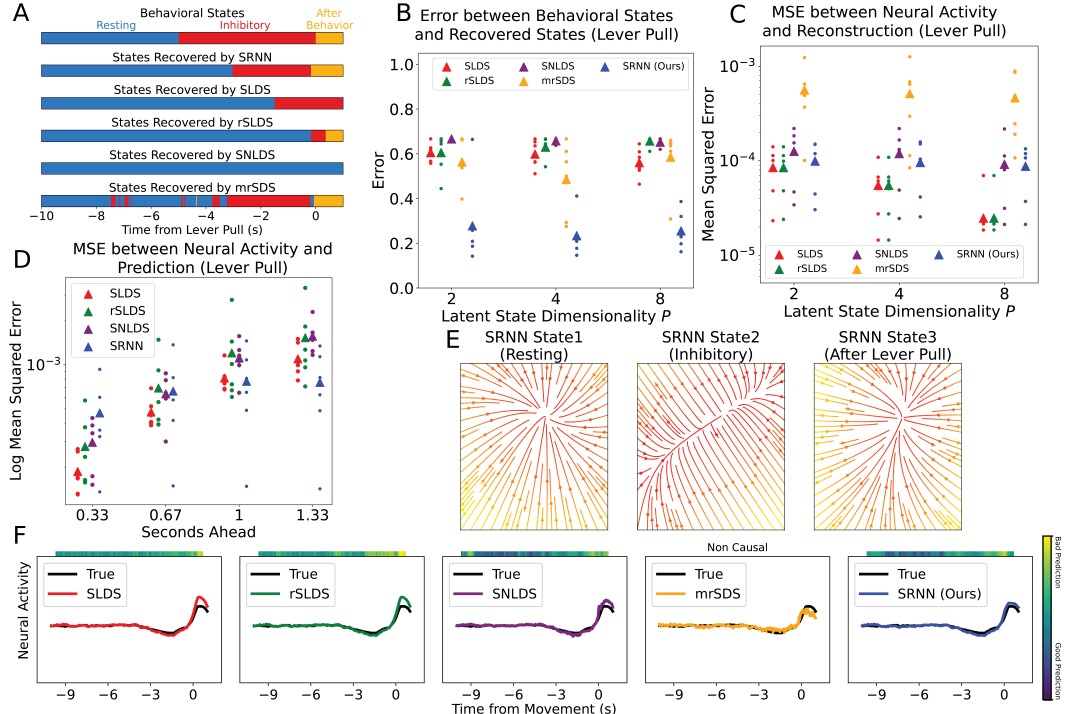

Figure 5: **WFCI Self-Initiated Lever Pull**: (A-F) Legend similar to that of Figure 4.

Moreover, the reconstruction of neural activity using SRNNs performs slightly worse than other methods (Figure 5C); however, the prediction of neural activity even 1 or 1.33 seconds ahead is better than competing methods. We show the reconstruction of neural activity from one example brain region in Figure 5F, with more results shown in Figure D.2D. Here, the color bars represent the performance of neural prediction. Indeed, all three methods have acceptable reconstruction and SRNNs have better prediction capabilities. Lastly, we visualize the neural dynamics in Figure 5E using $P = 2$. We find that the latent dynamics are only different from the starting of the inhibitory dynamics to behavior onset, which may reflect the mechanism through with internal dynamics in the brain achieve movements, with primarily changes in preparatory neural dynamics (here the inhibitory state) leading to upcoming movements (Churchland et al. [2012]). Finally, we explore modular SRNNs constrained to the activity of different regions in order to elucidate region-to-region communication in Figure D.9.

## 5    Limitations and Conclusions

We developed a novel model for recovering neural dynamics and neural state changes, termed Switching Recurrent Neural Networks (SRNNs). The results show that SRNNs outperform competing models in both recovering behaviorally-relevant neural states, and predicting future neural activity. However, certain limitations of this method still exist. Firstly, we set the maximum number of discrete states manually based on our observations of behavior, but we note that the reconstruction accuracy does not change drastically if we change this hyperparameter. Secondly, analogous to other generative models trained via Variational Inference, the model needs a good initialization to be trained efficiently; without this, the model may get stuck in one state and not adequately capture the switching nature of neural dynamics. Lastly, the complexity of the model increases the training time, for example, one experiment for SRNNs trained on reaching datasets may take several hours (e.g., around 6.5 hours on one session of curved reaching on NVIDIA A100 GPUs), which is more than the amount of time taken by SLDS and rSLDS (typically less than one hour). In conclusion, we use SRNNs to identify behaviorally-relevant neural dynamical states. We find that the neural dynamics have behaviorally-relevant switches across time and we are able to use SRNNs to capture these switches as well as the corresponding dynamical features. Straightforward extensions of this method include adding inputs or control signals to SRNNs to explore the identification of behaviorally-relevant neural dynamics while providing external stimuli.

While this paper focuses on neuroscience impact, methods such as SRNNs have the potential to be applied in health-related fields for positive societal impact towards design of intervention based on accurate prediction of neural dynamics. No negative societal impact is noted.

## Acknowledgement

We gratefully acknowledge support from the NIH Brain Initiative 7RF1DA056377-02 and NSF NCS 2350329. We thank Mark Churchland, Anne Churchland, Catalin Mitelut, and Timothy H Murphy for sharing experimental datasets. We also thank Nancy Padilla-Coreano for helpful discussions.

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

# A Simulated data: Lorenz Attractor

The equations of Lorenz Systems are:

$$\frac{dx}{dt} = \sigma(y - x) \tag{16}$$

$$\frac{dy}{dt} = x(\rho - z) - y \tag{17}$$

$$\frac{dz}{dt} = xy - \beta z \tag{18}$$

where $x, y, z$ represent 3 dimensions of the system, and $\sigma, \rho, \beta$ are three parameters. Here we use commonly used values for the parameters, i.e., $\sigma = 10, \rho = 28, \beta = 2.667$.

# B Experimental Datasets

**Electrophysiology Recordings during a Monkey Reaching Task**   In the reaching dataset, the recording starts with the monkey in a resting state, and a target shows up after 400ms. The monkey then gets a 'go' cue after a delay. Thereafter, the monkey starts moving from the origin to the targets and hold at the targets upon reaching. More details can be found in Churchland et al. [2012]. We explore the relationship between the change of neural dynamics and the change of behavioral states. Here, we use the trial-average neural activity which is averaged across multiple trials with the same conditions, details of the trial-average data can be found in Churchland et al. [2012]. The data is recorded for 180 neurons and has 236 time points representing 2.36 seconds.

**Widefield Calcium Imaging during Mouse Decision-Making Behavior**   In self-initiated decision making dataset, mice initiated trials by touching either of two handles and hold a handle for $1s$ followed by sensory stimuli. The sensory stimulus was presented for 600ms, and after a 500ms pause with no stimulus and then the stimulus was repeated for another 600ms. After the second stimulus, a 1000ms delay was imposed, then the mice are required to lick one of two spouts, the mice were rewarded with a drop of water if they licked the spout twice on the same side as the stimulus. More details can be found in Musall et al. [2019]. Here, we average across multiple trials recorded from the same mouse to create pseudo-sessions. Specifically, we averaged around 80 trials of 516 trials in one mice to produce $N = 8$ pseudo-sessions, and fit the trial-averaged data using our models. The details of the data can be found in Musall et al. [2019]. The data is also processed via LocaNMF Saxena et al. [2020], and the 210 temporal components returned by LocaNMF are used. The data has 189 time points representing 6.3 seconds.

**Widefield Calcium Imaging during Mouse Self-initiated Lever Pull Behavior**   In self-initiated lever pull data, mice were trained to pull a lever and hold it at an angle (for $> 100$ms) in order to receive a water supplement. We then apply LocaNMF Saxena et al. [2020] while spatially aligning the imaged neural activity with the Allen mouse brain coordinate framework Wang et al. [2020] using affine transformations, as previously performed in Musall et al. [2019], and take 16 components as identified by LocaNMF, which form our input signals, with each input dimension from one brain region. In this work, we focus on the signals around the lever pull, i.e., from 10 seconds before lever pull to 1 second after lever pull (overall 330 time points). More details can be found in Mitelut et al. [2022]. Likewise, we use trial-averaged data which is averaged across all trials of the same mouse. We use the data across $N = 6$ mice; details of this data can be found in Mitelut et al. [2022].

# C Region-to-region Communication on WFCI

We also use our model to uncover the relationship between different brain regions using self-initiated lever pull dataset recorded by WFCI, because the data has only one component for each brain region. We add a constraint to our SRNNs, specifically, we force 4 hidden states of SRNNs to reconstruct neural activity from 4 brain region independently via constraining the output weights of SRNNs. We focus on communication between 'Motor', 'Visual', 'Somatosensory', and 'Other' regions except these 3. Finally, we visualize the recurrent weights of each RNN of SRNNs in Figure D.9A, we find that the region-to-region communication is highly different between different states. Moreover, the

self-communication during rest state (diagonal of SRNN1) is at a low amplitude. Nevertheless, this self-communication becomes higher during the 'inhibitory dynamics' (SRNN2), then decreases in 'after lever pull' (SRNN3). Additionally, the 'Visual' region has more communication with other regions during the states prior to the behavior onset (SRNN1 and SRNN2), the 'Somatosensory' region has little communication at early states, but it then increases close to behavior onset. The 'Motor' region actively communicates with other regions near behavior.

# D  Appendix Figures

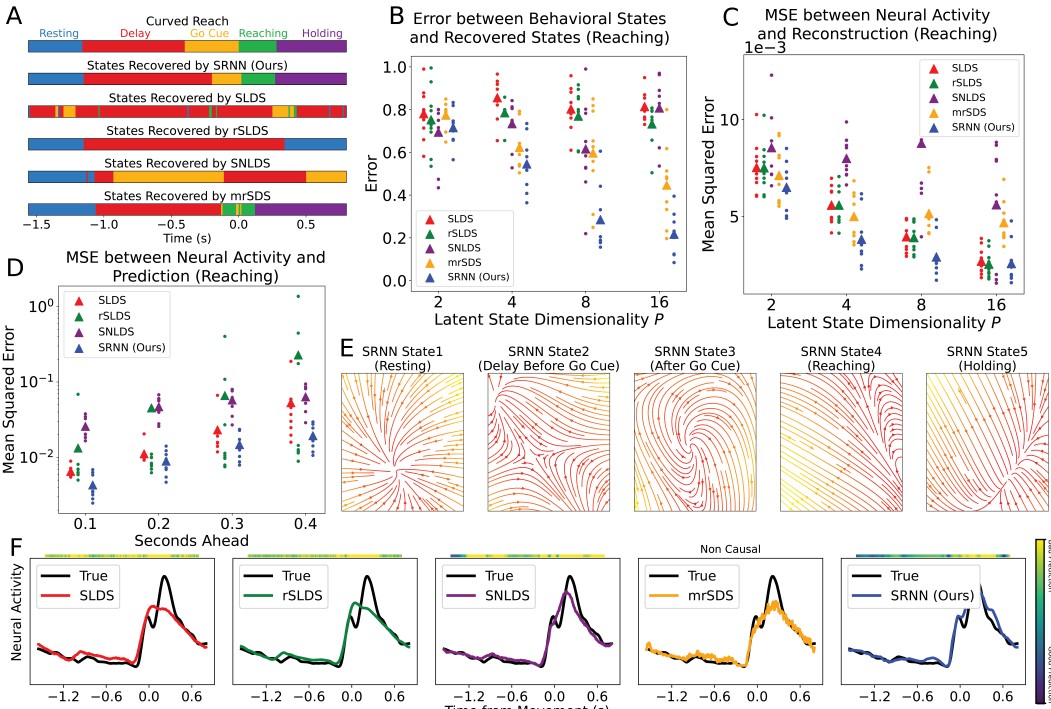

Figure D.1: **Direct Reaching**: (A) Examples of comparison between behavioral states and neural dynamical states recovered by SRNNs, SLDS, and rSLDS. (B) Error between behaviorally-relevant states and recovered neural dynamical states; each dot represents one session, with the triangle as the mean of all sessions. (C) Mean squared error (MSE) between neural activity and the reconstruction. (D) MSE between neural activity and the *prediction* forward in time. (E) Reconstruction of neural activity and the corresponding ground truth for one example neuron for three models, the top color bars represent the performance on $0.1$ seconds ahead prediction. More examples are shown in Figure D.2. (F) Flow fields of latent dynamics of SRNNs in different discrete states.

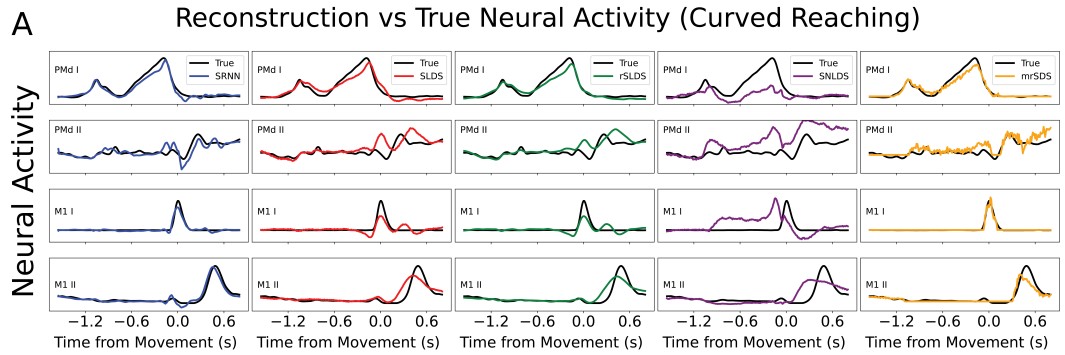

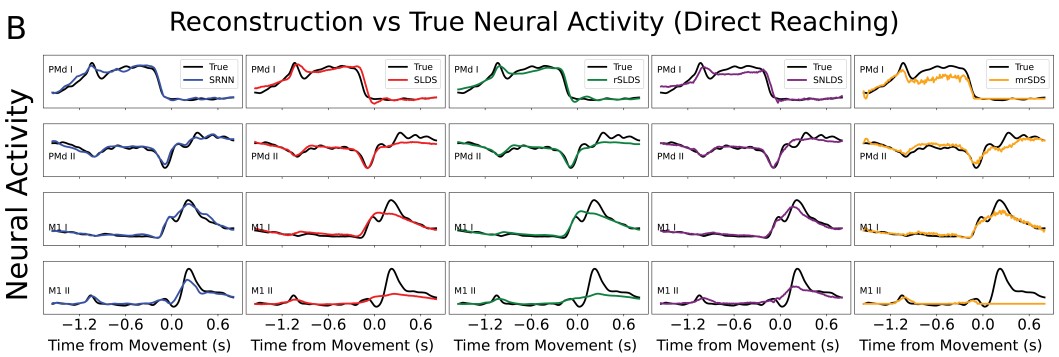

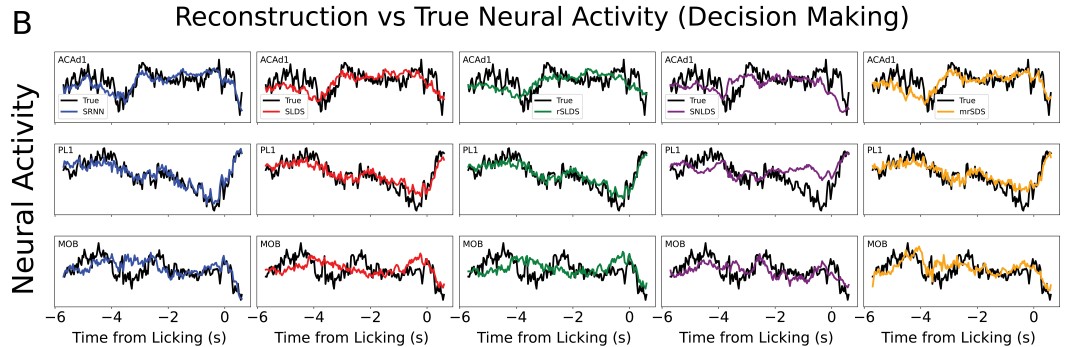

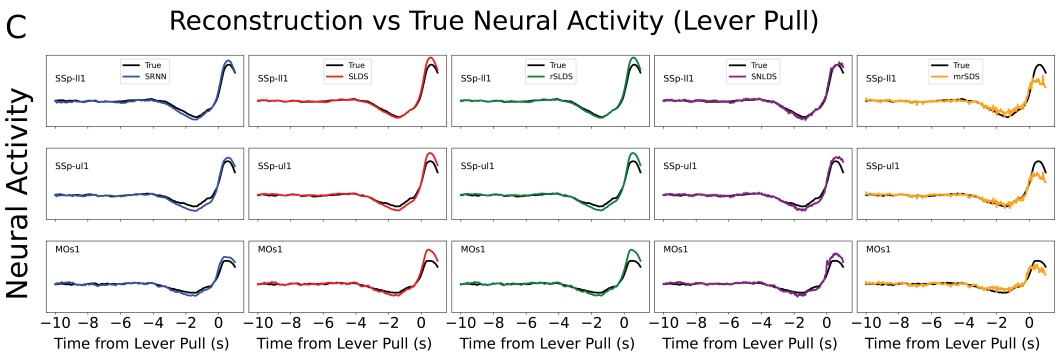

Figure D.2: Reconstruction of neural activity and the corresponding ground truth in different brain region/neuron for (A) curved reaching, (B) direct reaching, (C) self-initiated decision making, and (D) self-initiated lever pull. Red color represents SLDS, green color represents rSLDS, blue color represents SRNN, purple color represents SNLDS, and orange color represents mrSDS.

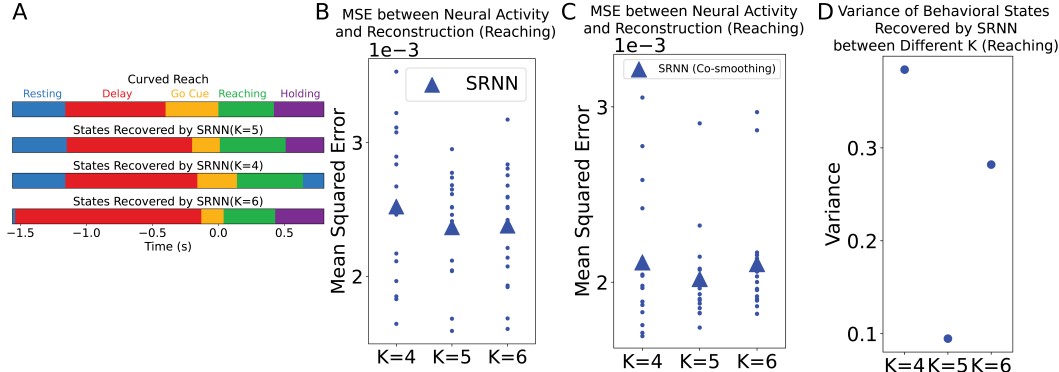

Figure D.3: **Curved Reaching**: (A) Examples of comparison between behavioral states and neural dynamical states recovered by SRNN with $K = 4$, $K = 5$, and $K = 6$. (B) Mean squared error (MSE) between neural activity and the reconstruction by SRNN with $K = 4$, $K = 5$, and $K = 6$. (C) Mean squared error (MSE) between neural activity and the reconstruction by SRNN with $K = 4$, $K = 5$, and $K = 6$ using the co-smoothing test. (D) Variance of behavioral states recovered by SRNN with $K = 4$, $K = 5$, and $K = 6$.

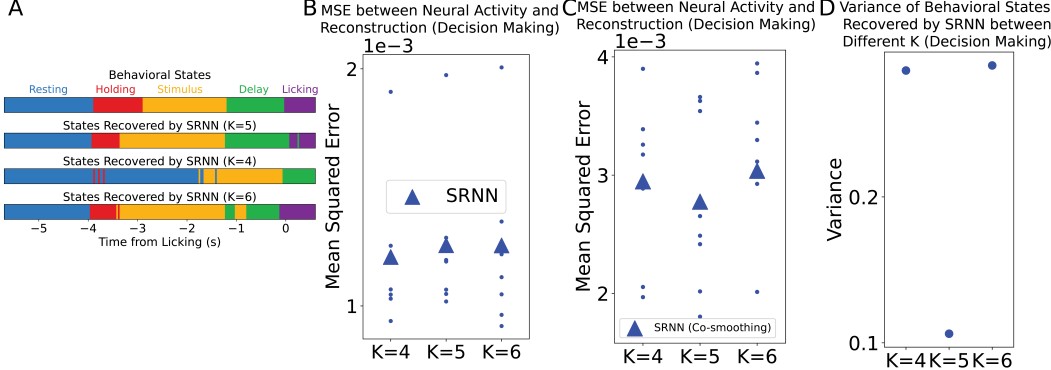

Figure D.4: **WFCI Decision-Making**: (A) Examples of comparison between behavioral states and neural dynamical states recovered by SRNN with $K = 4$, $K = 5$, and $K = 6$. (B) Mean squared error (MSE) between neural activity and the reconstruction by SRNN with $K = 4$, $K = 5$, and $K = 6$. (C) Mean squared error (MSE) between neural activity and the reconstruction by SRNN with $K = 4$, $K = 5$, and $K = 6$ using the co-smoothing test. (D) Variance of behavioral states recovered by SRNN with $K = 4$, $K = 5$, and $K = 6$.

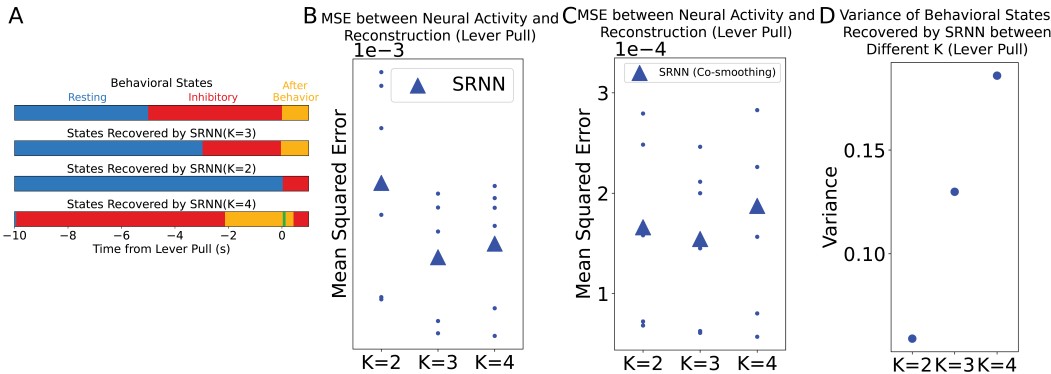

Figure D.5: **WFCI Self-Initiated Lever Pull**: (A) Examples of comparison between behavioral states and neural dynamical states recovered by SRNN with $K = 2$, $K = 3$, and $K = 4$. (B) Mean squared error (MSE) between neural activity and the reconstruction by SRNN with $K = 2$, $K = 3$, and $K = 4$. (C) Mean squared error (MSE) between neural activity and the reconstruction by SRNN with $K = 2$, $K = 3$, and $K = 4$ using the co-smoothing test. (D) Variance of behavioral states recovered by SRNN with $K = 2$, $K = 3$, and $K = 4$.

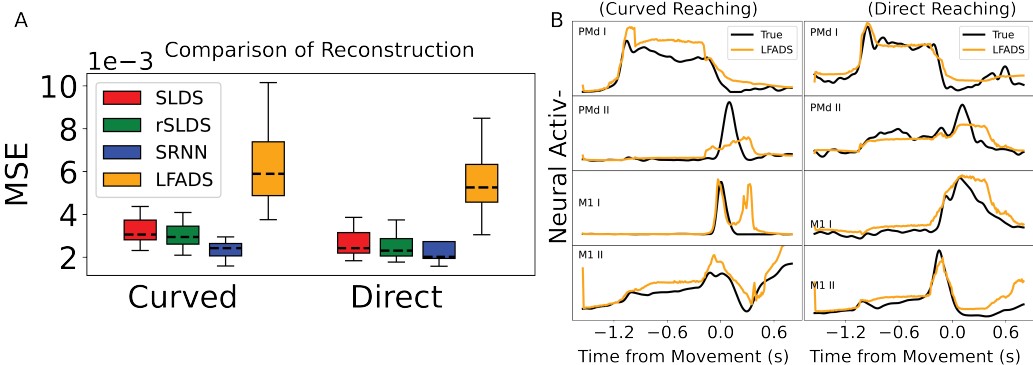

Figure D.6: (A) Comparison of neural activity reconstruction among SLDS, rSLDS, SRNN and LFADS in reaching dataset. (B) Reconstruction of neural activity and the corresponding ground truth in four example neurons for LFADS.

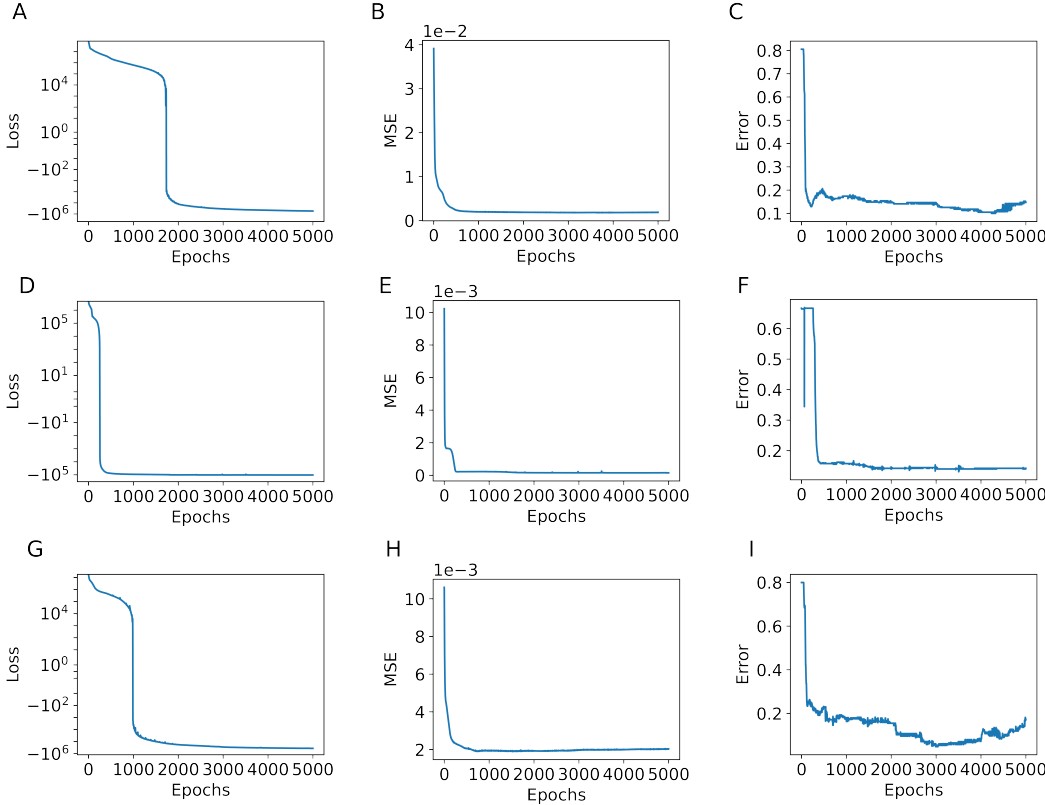

Figure D.7: First column: Plot of SRNN training loss across 5000 epochs for example run in (A) reaching dataset, (D) self-initiated lever pull dataset, and (G) self-initiated decision making dataset. Second column: Plot of MSE between desired neural activity in validation set and the corresponding reconstruction by SRNN for (B) reaching dataset, (E) self-initiated lever pull dataset, and (H) self-initiated decision making dataset. Third column: Plot of error between behavioral states in validation set and the corresponding recovered state by SRNN for (C) reaching dataset, (F) self-initiated lever pull dataset, and (I) self-initiated decision making dataset.

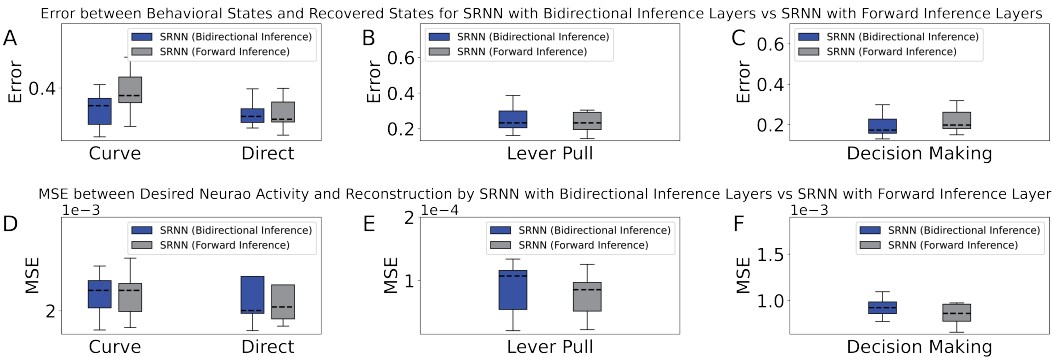

Figure D.8: Error between behavioral states and recovered states by SRNN with bidirectional recurrent inference layers (blue) vs SRNN with forward recurrent inference layers (gray) in (A) reaching dataset, (B) self-initiated lever pull dataset, and (C) self-initiated decision making dataset. MSE between desired neural activity and reconstruction by SRNN with bidirectional recurrent inference layers (blue) vs SRNN with forward recurrent inference layers (gray) in (D) reaching dataset, (E) self-initiated lever pull dataset, and (F) self-initiated decision making dataset.

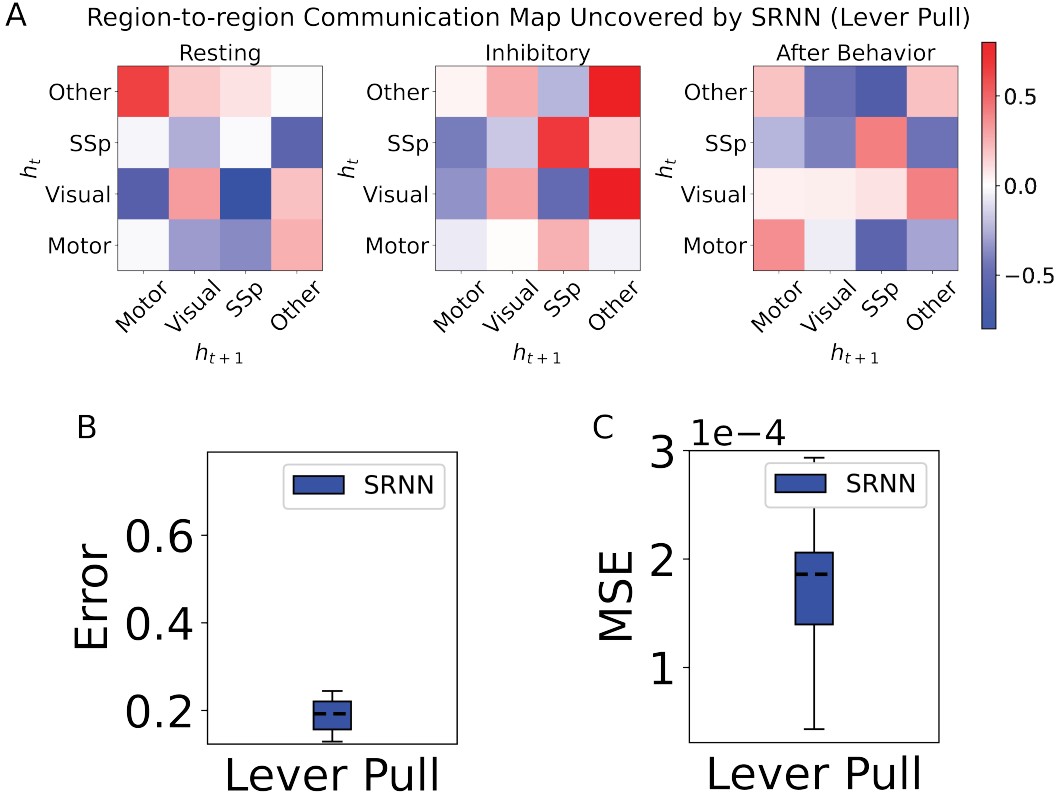

Figure D.9: (A) Region-to-region communication map uncovered by constrained SRNN trained on self-initiated lever pull dataset. We focus on 4 regions, i.e., "Visual", "Motor", "Somatosensory (SSp)", and "Others". (B) Error between behavioral states and recovered states by constrained SRNN on self-initiated lever pull dataset. (C) MSE between desired neural activity and reconstruction by constrained SRNN on self-initiated lever pull dataset.

