# OpenReview forum: "Inference of Neural Dynamics Using Switching Recurrent Neural Networks"
_NeurIPS.cc/2024/Conference — NeurIPS 2024 poster_

### Official Review · Reviewer_uaif · 2024-07-07

**Soundness:** 3
**Presentation:** 2
**Contribution:** 2
**Rating:** 5
**Confidence:** 4

**Summary:**

This paper develops a switching RNN (SRNN) framework to model neural activity. It builds up on switching linear dynamical system models that are used in neuroscience to segment and extract underlying dynamics of observed neural activity. The different segments corresponding to unique dynamics often reflect distinct behavioral states. The crucial novelty of this work is that they allow the dynamics to be non-linear, unlike SLDS and rSLDS, making the model more expressive. They fit these models using VI using an inference network. Finally, they apply SRNN to synthetic data, as well as 3 distinct neural datasets and show that it outperforms SLDS and rSLDS on segmenting activity into behavioral modules where each module corresponds to distinct dynamics. They visualize these underlying dynamics, and also evaluate their fitted model on predicting future neural activity.

**Strengths:**

1. As we move towards large-scale neural datasets, it is crucial to scale model complexity in order to fully harness these datasets. This paper makes a step in that direction by allowing for non-linear dynamics, while also providing an appropriate fitting approach.
2. The experiment section is extensive, and I appreciate the application to multiple neural datasets. I particularly found the results on the decision-making dataset to be most impressive.
3. The literature review is thorough, and the authors do a good job of situating their work in the context of other related studies.

**Weaknesses:**

1. The authors mention switching nonlinear dynamical systems (Dong et al. 2020), and discuss how their work differs from Dong et al. I think it is important to either provide an experimental comparison to SNLDS or a justification for why these existing models are insufficient to explain neural datasets, as the main novelty/motivation for SRNN and SNLDS is very much related (also noted by the authors in the paper). More on this in the  question section.

2. Behavioral segmentations are somewhat subjective in nature, and while I can see that in the experiments shown here they make sense, in a real world setup we may want to infer the number of such segmentations from the data. Here the authors set the number of discrete states to the # of true behavioral states, however this might not be known in practice. Furthermore, there might be distinct sets of dynamics within one behavioral state due to other reasons not totally explicit from behavior. From the current set of results, it is not clear if SRNN is capable of inferring the # of underlying states. I will elaborate more in the questions section on this as well.

3. I also think the paper will benefit from some editing by the authors. The references are not formatted properly, commas are missing. The referencing to supplementary figures doesn't seem to be working, it links back to figures in the main text. I also think the authors can trim some of the background, such as the section on VI, in favor of explaining some of the experiments such as the Lorenz attractor setup in more detail.

4. While I appreciate the extensive experiments, I find it hard to reconcile some of the results. It seems like in some of the plots (Fig 3C/D, Fig 5C/D) prediction + reconstruction performance across all models is similar. However, the discrete states being inferred look hugely inaccurate for SLDS and rSLDS. I wonder if the authors have thoughts on why this happens.

**Questions:**

1. I feel it is crucial to understand the pros and cons of SNLDS vs SRNN, and understand if the differences between the two models are consequential in modeling neural data. For example, I understand some of the modeling differences such as the dependence of discrete state transitions on the data in SNLDS vs the previous continuous state in this paper, but I am not sure if one of the two is a better assumption. Additionally, I am also struggling to understand why can SNLDS not be used for prediction?

2. In the real-world experiments, were the number of discrete states across all models being compared set to the true number of behavioral states? I would be curious to see how results vary across different # of discrete states across these models, perhaps MSE on prediction or ELBO vs # of discrete states is a possible way to show this. This is for two mains reasons:
i. In a new dataset we might not know ground-truth behavioral segmentations, and will want to be able to infer the number of such segmentations from the data. Hence, it would be interesting to see if SRNN can be used to do so.
ii. The fact that rSLDS does fine it predicting data but infers discrete states inaccurately makes me wonder if it is clustering data differently, perhaps collapsing 2 states into one or further segmenting one behavioral state into multiple slightly different states.

**Limitations:**

The authors have addressed limitations in the last section of the paper, and I do not envision any societal impact of this work.

---

> ### Author Rebuttal · Authors · 2024-08-07
>
> We thank you deeply for your time and attention in reading our paper, and for your valuable comments. Below is our response to specific weaknesses and questions.
>
> **Summary of Weakness 1**: *comparison to SNLDS.*
>
> We thank you for pointing out this weakness. In addition to the existing SLDS, rSLDS, and LFADS, we implemented two additional baseline models that reviewers mentioned, the SNLDS [Dong et al., 2020] and mrSDS [Karniol-Tambour et al., ICLR 2024]. We show their performance on the Reaching Dataset in the attached PDF. To summarize, while recovering behaviorally-relevant states, our model SRNN outperforms SNLDS and has comparable performance with mrSDS, whereas in reconstructing neural activity, SRNN outperforms mrSDS and has comparable performance with SNLDS . Therefore, SRNNs have overall better performance on recovering behaviorally-relevant states and reconstructing neural activity.
>
> **Q1)**: *It is crucial to understand the pros and cons of SNLDS vs SRNN, and understand if the differences between the two models are consequential in modeling neural data. For example, I understand some of the modeling differences such as the dependence of discrete state transitions on the data in SNLDS vs the previous continuous state in this paper, but I am not sure if one of the two is a better assumption. Additionally, I am also struggling to understand why can SNLDS not be used for prediction?*
>
> We thank you for allowing us to elaborate on the differences between the two models. (1) Indeed, the transitions in our model are inspired by rSLDS, i.e., in SRNNs, the transition depends on continuous states $h_t$. However, we observed no significant performance difference between transitions based on the observations $y_t$ as compared to $h_t$, as $y_t$ is a linear transformation of $h_t$. However, $h$ typically has a lower dimensionality, requiring fewer parameters to model the transition, e.g., in the context of reaching, the dimensions of the continuous latent variables, $h$, P=16, vs. the dimensions of neural activity, $y$, R=180. (2) SNLDS is able to be tweaked in order to make predictions by using a causal inference network, which is the approach utilized in our prediction work. The original SNLDS paper used non-causal inference, which involves inferring states from the entire sequence. Consequently, the original SNLDS method is unsuitable for predicting neural activity, as it requires information from future neural activity or even the entire neural activity sequence for state inference, but again, this is able to be tweaked for the purpose of prediction. (3) Lastly, SNLDS and SRNNs have different initialization processes, which we suspect is contributing to SRNNs’ better performance in modeling neural data (see response to Weakness 1). While SNLDS uses regularization in the objective function to ensure uniform state utilization, SRNNs use HMM as an initialization, again inspired by previous models that have been successfully applied towards modeling neural data.
>
> **Summary of Weakness 2**: *It is not clear if SRNN is capable of inferring the # of underlying states.*
>
> **Summary of Question 2**: *how results vary across different # of discrete states across all models. This is for two main reasons: i. In a new dataset we might not know ground-truth behavioral segmentations. ii. The fact that rSLDS does fine in predicting data but infers discrete states inaccurately.*
>
> We thank you for your questions. We now provide a comprehensive set of metrics to determine this important hyperparameter in the **global rebuttal** as well as in the official comment below. To summarize, we use appropriate selection metrics for the number of discrete states: (1) convergence / reuse of discrete states, (2) reconstruction performance and co-smoothing method, and (3) variability across conditions or trials. We show comprehensive results for these metrics on two experimental datasets where we recover the appropriate number of behaviorally relevant states using any of these three metrics.
>
> Moreover, we also show reconstruction performance across all models with different K in the attached PDF (Figure 2).
>
> Lastly, we agree that rSLDS does fine in reconstruction of data but infers discrete states inaccurately. Please see below (response to Weakness 4) for a discussion on this topic.
>
> **Summary of Weakness 3**: *The paper will benefit from some editing by the authors.*
>
> We thank you for pointing out this weakness and we apologize for the inconvenience in locating the figures when you review our paper. We have fixed the reference formatting and the figure references. Additionally, we have added some simulation and experimental details in the main text, and removed some commonly-used VI details to the Appendix.
>
> **Summary of Weakness 4**: *Prediction + reconstruction performance across all models is similar. However, the discrete states being inferred look hugely inaccurate for SLDS and rSLDS.*
>
> We thank you for pointing this out; this has puzzled us as well. We believe that accurate reconstruction of data does not necessitate accurate inference of discrete hidden states or dynamics, since the models are trained purely for maximizing the ELBO, i.e., here, reconstruction accuracy. However, accurate prediction of neural activity many time steps ahead requires simultaneously (a) accurate reconstruction, (b) accurate inference of discrete states, and (c) accurate inference of dynamics. Here, we see concrete differences in the prediction capability of SRNNs vs. SLDS / rSLDS (e.g., see Figure 4D of the original submission). We believe that prediction of future activity provides a better test of a dynamical model, and here, SRNNs have overall better performance in prediction of neural activity many time steps ahead.

---

> ### Author Response · Authors · 2024-08-07
>
> **Details of Determining relevant hyperparameters (Weakness 2 and Question 2)**
>
> We now add a discussion for determining relevant hyperparameters, **such as the number of discrete and continuous latent states**. Importantly, we include a comprehensive hyperparameter sweep for the number of discrete states for both the reaching and decision-making datasets. We detail out the salient results for the reaching dataset, where there are 5 behavioral states in the task (decision-making results below):
>
> (1) **Convergence to lower number of discrete states**: We tested our model by increasing the number of hidden states K while keeping the number of continuous latent states P constant. We found that **61%** of SRNNs with a higher number of discrete hidden states (e.g., K=6) finally converge to the optimal number of discrete hidden states, i.e., K=5.
>
> (2) **Reuse of discrete states**: We also test our model by decreasing the number of hidden states. We found that **94%** of SRNNs with a lower number of discrete hidden states (e.g., K=4) had at least one hidden state reused after other states: in other words, SRNNs are not able to perform well with 4 unique discrete hidden states without reusing one of them.
>
> (3) **Reconstruction performance plateau**: While keeping other hyperparameters constant, the reconstruction accuracy plateaus at the same number of discrete states as in the behavior, thus we can use the minimum number of discrete states as it takes for the model to perform well. We have included a figure detailing this in the attached PDF (Figure 2). Moreover, we also implemented a ‘co-smoothing’ method as suggested by Reviewer hFy4 [Yu et al., 2009 and Karniol-Tambour et al., ICLR 2024], we show the results in the attached PDF (Figure 3), where we found that K=5 also does well in reconstructing the data with a ‘co-smoothing’ neuron drop-out analysis.
>
> (4) **Variability across conditions**: In stereotyped tasks or experiments, such as reaching, there may not be a significant amount of variability in the timing of behavior across conditions, and this variability can thus be used as a metric for determining the number of discrete states. **Here, we found that SRNNs with K=5 have much lower variability on recovered behaviorally-relevant states than K=4 and K=6 (i.e., 0.098 for K=5, 0.384 for K=4, and 0.282 for K=6).**
>
> Furthermore, we did the same analysis for the decision-making data (Figure 4 in the attached PDF) and we found the same situation as in the reaching dataset. **Higher K also converges to a smaller number of discrete states, and K=5 has the smallest variability across different pseudo-sessions (i.e., 0.106 for K=5, 0.287 for K=4, and 0.290 for K=6).**
>
> These results demonstrate appropriate selection metrics for the number of discrete states: (1) convergence / reuse of discrete states, (2) reconstruction performance, and (3) variability across conditions or trials. We show comprehensive results for these metrics on two experimental datasets where we recover the appropriate number of behaviorally relevant states using any of these three metrics. We have now provided this in the paper, while adding a discussion for the general case.
>
> Additionally, we show a comparison between values for another important hyperparameter P in the original submission.

---

> ### Comment · Reviewer_uaif · 2024-08-11
> **Response to rebuttal**
>
> Thank you for your detailed response. I appreciate the new results: comparisons with SNLDS and mrSDS, as well as the detailed experiments on discrete state switching. I still find myself a bit perplexed by the discrepancy in behavior and neural data results and I wish I understood the advantages / flaws of each of these models better. Overall, since the authors have addressed some of my concerns, I will raise my score to beyond the accept threshold.

---

> > ### Author Response · Authors · 2024-08-12
> >
> > We thank you for your response and for your consideration in raising the score. We didn’t find this discrepancy in recovering behavioral vs. reconstructing neural data in our model. We definitely agree that this discrepancy in existing models is interesting to explore.

---

### Official Review · Reviewer_Kpfw · 2024-07-11

**Soundness:** 3
**Presentation:** 3
**Contribution:** 3
**Rating:** 6
**Confidence:** 4

**Summary:**

The authors develop a new class of probabilistic nonlinear state space models called switching RNNs. In essence, this extends the well-known switching linear dynamical system (SLDS) model to switch between nonlinear dynamics governed by a stochastic RNN.

**Strengths:**

* The results shown in panels A of Figs 3, 4, and 5 are nice and convincing.

**Weaknesses:**

* Like many other deep learning based approaches, the model is not particularly interpretable. For example, panel F in Figs 3, 4, and 5 shows 2D flow fields for the different hidden states, but the RNN hidden state is 16-dimensional. Here the authors have used PCA to attempt to find a reasonable 2D flow field, but I know from experience that this has the potential to very poorly capture the true dynamics of the system. Intuitively, even small variance dimensions can matter a lot if the flow field changes rapidly along that dimension.

* There are many tunable parameters in this model (e.g. number of continuous and number of discrete states). It is unclear how to choose these on datasets without ground truth, or at least good educated guesses.

* Related to above, I worry a lot about the identifiability of this model. A nonlinear RNN without discrete switching can already model any flow field if given enough units. Thus a model with many continuous states (e.g. $P=128$) but zero discrete states may perform equally well to a model with few continuous states (e.g. $P=16$ or $P=8$) but a handful of discrete states. How would one then go about choosing between these models? Adding discussion or ideally some sort of mathematical analysis regarding the statistical identifiability of the model would be very helpful.

**Questions:**

* Equation (2) seems wrong to me. The nonlinearity $f(\cdot) = \tanh(\cdot)$ doesn't make sense here since $p(z_t \mid z_{t-1}, h_{t-1})$ should be a positive number. Perhaps you meant to use a softmax nonlinearity here?

* Equation (2) also seems to suggest that the transition probability only depends on $h_{t-1}$ and not $z_{t-1}$. Is this correct?

* Related to the above, it wasn't obvious to me whether you are allowing the continuous hidden state to impact the transition for the discrete state. Essentially I am wondering if your model is analogous to the switching LDS (where the continuous hidden state doesn't impact the transition statistics) or instead the recurrent switching LDS (where the continuous state does impact the discrete transition probabilities). In Figure 1B, what is the meaning of the red dashed arrow? Does that carry any difference to the black arrows?

* Are the good results shown in panel A of Figs 3, 4, and 5 due solely to differences in the initialization procedure across models? (see top of page 5)

* Regarding identifiability, what happens if you run the model multiple times from different random seeds? Do you recover the same flow fields and fixed point structure?

**Limitations:**

The discussion adequately acknowledges limitations.

---

> ### Author Rebuttal · Authors · 2024-08-07
>
> We thank you very much for giving us a positive rating, and for your extremely relevant  comments. Below we respond to specific weaknesses pointed out.
>
> **Summary of Weakness 1**: *Like many other deep learning based approaches, the model is not particularly interpretable. 2D flow fields poorly capture the true dynamics of the system.*
>
> Yes, you are absolutely correct, a 2D flow field is not able to accurately capture the true dynamics when P>2, and simply provides a visualization. We are able to also calculate slow points and other features of the dynamics in the full dimensional space, provided here as an example in Figure 8 of the attached PDF.
> When comparing the dynamics in different discrete states of our model, we use the same principal components to project the higher-dimensional dynamics to 2D. Therefore, the dynamics can technically be compared with each other. Additionally, where possible (e.g., in Figure 5F of the original submission), we plot the flow fields for SRNNs with P=2; here, the visualization and structure found is of the true dynamics. We have also run this model with a different random seed and recovered similar features of the dynamics in Figure 8 of the attached PDF, showing identifiability of the dynamics.
>
> **Summary of Weakness 2**: *It is unclear how to choose tunable parameters on datasets without ground truth, or at least good educated guesses.*
>
> We thank you for your comments, and comprehensively discuss this in the **global rebuttal** as well as in the official comment below. To summarize, we use appropriate selection metrics for the number of discrete states: (1) convergence / reuse of discrete states, (2) reconstruction performance and co-smoothing method, and (3) variability across conditions or trials. We show comprehensive results for these metrics on two experimental datasets where we recover the appropriate number of behaviorally relevant states using any of these three metrics.
>
> Additionally, for selecting the number of continuous hidden states P, we included comparisons in the paper, we think an appropriate P should perform well on both recovering states constantly and reconstructing behavior, meanwhile, the P should be as small as possible.
>
> **Summary of Weakness 3**: *How to choose between a SRNN and a model with higher P.*
>
> Yes, you are definitely right. According to the theory of universal function approximation, a model with higher dimensions can do the same work as multiple models with low dimensions. To empirically test this, we trained a standard RNN with P=32, which has a similar number of parameters on recurrent weights but more on emission weights, and found that standard RNNs are also able to reconstruct neural activity well (see attached PDF Figure 7). However, neural dynamics are broadly thought to be low-dimensional, and we would like this dimensionality to be as low as possible to achieve interpretability in the dynamics as well as the discrete states (also discussed above). Indeed, we also found that the switches in the neural dynamics are  relevant to different behavioral and stimulus states; therefore, if we use a higher dimensionality but a non-switching model, the model may not capture this interpretability.
>
> **Q1)**: *Typo in Equation (2)*
>
> We thank you for pointing this out, it is a typo, and we used softmax in implementation. We will revise the equation.
>
> **Q2 and Q3)**: *It wasn't obvious to me whether you are allowing the continuous hidden state to impact the transition for the discrete state. In Figure 1B, what is the meaning of the red dashed arrow? Does that carry any difference to the black arrows?*
>
> We thank you for your question. The value of the transition probability depends on $h_t$. Our model is similar to rSLDS, where the transition probabilities depend on continuous states. We plot the red dashed arrow to show that the transition probabilities depend on previous continuous states (they are red since the states z are not directly computed using $h_t$). The black arrows between z represent the transitions between z.
>
> **Q4)**: *Are the good results shown in panel A of Figs 3, 4, and 5 due solely to differences in the initialization procedure across models?*
>
> We thank you for your question. Initialization might be the most common and powerful method to overcome the difficulty of training in this area. We used the SSM package to implement SLDS and rSLDS where HMMs are also used as an initialization.
>
> **Q5)**: *Regarding identifiability, what happens if you run the model multiple times from different random seeds? Do you recover the same flow fields and fixed point structure?*
>
> We thank you for your question. We agree that we should see the same structure in the dynamics across different random seeds. To show a simple example, we changed the random seed and trained the SRNN on lever pull data with P=2. We show a visualization in the attached PDF in Figure 8. We found similar flow fields with a comparable fixed point structure. Specifically, the new fixed points seem to be rotated from the old ones in the same direction.

---

> ### Author Response · Authors · 2024-08-07
>
> **Details of Determining relevant hyperparameters (Weakness 2)**
>
> We now add a discussion for determining relevant hyperparameters, **such as the number of discrete and continuous latent states**. Importantly, we include a comprehensive hyperparameter sweep for the number of discrete states for both the reaching and decision-making datasets. We detail out the salient results for the reaching dataset, where there are 5 behavioral states in the task (decision-making results below):
>
> (1) **Convergence to lower number of discrete states**: We tested our model by increasing the number of hidden states K while keeping the number of continuous latent states P constant. We found that **61%** of SRNNs with a higher number of discrete hidden states (e.g., K=6) finally converge to the optimal number of discrete hidden states, i.e., K=5.
>
> (2) **Reuse of discrete states**: We also test our model by decreasing the number of hidden states. We found that **94%** of SRNNs with a lower number of discrete hidden states (e.g., K=4) had at least one hidden state reused after other states: in other words, SRNNs are not able to perform well with 4 unique discrete hidden states without reusing one of them.
>
> (3) **Reconstruction performance plateau**: While keeping other hyperparameters constant, the reconstruction accuracy plateaus at the same number of discrete states as in the behavior, thus we can use the minimum number of discrete states as it takes for the model to perform well. We have included a figure detailing this in the attached PDF (Figure 2). Moreover, we also implemented a ‘co-smoothing’ method as suggested by Reviewer hFy4 [Yu et al., 2009 and Karniol-Tambour et al., ICLR 2024], we show the results in the attached PDF (Figure 3), where we found that K=5 also does well in reconstructing the data with a ‘co-smoothing’ neuron drop-out analysis.
>
> (4) **Variability across conditions**: In stereotyped tasks or experiments, such as reaching, there may not be a significant amount of variability in the timing of behavior across conditions, and this variability can thus be used as a metric for determining the number of discrete states. **Here, we found that SRNNs with K=5 have much lower variability on recovered behaviorally-relevant states than K=4 and K=6 (i.e., 0.098 for K=5, 0.384 for K=4, and 0.282 for K=6).**
>
> Furthermore, we did the same analysis for the decision-making data (Figure 4 in the attached PDF) and we found the same situation as in the reaching dataset. **Higher K also converges to a smaller number of discrete states, and K=5 has the smallest variability across different pseudo-sessions (i.e., 0.106 for K=5, 0.287 for K=4, and 0.290 for K=6).**
>
> These results demonstrate appropriate selection metrics for the number of discrete states: (1) convergence / reuse of discrete states, (2) reconstruction performance, and (3) variability across conditions or trials. We show comprehensive results for these metrics on two experimental datasets where we recover the appropriate number of behaviorally relevant states using any of these three metrics. We have now provided this in the paper, while adding a discussion for the general case.
>
> Additionally, we show a comparison between values for another important hyperparameter P in the original submission.

---

> > ### Comment · Reviewer_Kpfw · 2024-08-09
> > **Reviewer Response**
> >
> > Thanks for the additional points and clarifications. I retain my score of a "weak accept" as I think the work is novel, technically correct, and could be of interest to the neural modeling community. I still have reservations about impact of this method on the broader neuroscience community, given challenges related to interpretability and somewhat weak reasons to prefer this method over a higher dimensional RNN.

---

> ### Author Response · Authors · 2024-08-10
>
> >*Thanks for the additional points and clarifications. I retain my score of a "weak accept" as I think the work is novel, technically correct, and could be of interest to the neural modeling community. I still have reservations about impact of this method on the broader neuroscience community, given challenges related to interpretability and somewhat weak reasons to prefer this method over a higher dimensional RNN.*
>
> We thank you for your response and retaining a positive rating. We agree on the trade-off between switching low-dimensional RNNs and non-switching higher-dimensional RNNs. If accuracy is the only metric, we agree that high-dimensional standard RNNs are able to achieve this goal. However, for achieving interpretability in (a) identifying discrete behaviorally-relevant switches in dynamics, and (b) distinctive flow-field visualization, we demonstrate the utility of switching RNNs in the paper. As a concrete example comparing the two models, in order to achieve similar reconstruction accuracy as a P=2, K=3 SRNN (lever-pull data), we need a standard RNN with dimensionality P=5. This higher dimensionality in the standard RNN naturally comes at the cost of accuracy in the 2D flow-field representations, and thus may result in compromised interpretability of the dynamics.
>
> The trade-offs between low-dimensional switching RNNs and higher-dimensional standard RNNs are interesting, and we will add quantitative accuracy tradeoffs with varying values for P and K in the final version, if accepted.

---

### Official Review · Reviewer_hFy4 · 2024-07-12

**Soundness:** 2
**Presentation:** 2
**Contribution:** 2
**Rating:** 3
**Confidence:** 4

**Summary:**

The authors propose to model time series neural population activity using switching recurrent neural networks. The generative model includes discrete latent states

**Strengths:**

The proposed method does appear to outperform related switching linear dynamical systems approaches in certain contexts.

**Weaknesses:**

High-level:
- The contribution beyond other switching nonlinear dynamical systems models is not clear. Such models include the cited Dong et al., 2020, as well as Karniol-Tambour et al., ICLR 2024. If there is a contribution beyond these works, the authors should compare against those existing related methods.
- The authors do not demonstrate an ability to automatically determine the appropriate number of discrete states. One approach to this might be "co-smoothing" (see Yu et al., Gaussian Process Factor Analysis, 2009).

Details:
- The mathematical details and notation are often unclear. For example, equation 2 does not appear to be a valid probability distribution, given the description that f(.) = tanh(.). Shouldn't this instead be a categorical distribution or similar? Relatedly, f is also used in equation 8, but from the context it appears to denote something entirely different.
- The authors should more clearly describe the cross-validation techniques for used for each dataset. The blanket statement in the intro to Section 4 ("On each dataset, we do N-fold cross-validation, where N equals to the number of conditions, sessions, or subjects in the dataset") obscures how cross-validation was actually applied in each instance.

**Questions:**

- Are the predictions in Figure 2 cross validated (eg., using the technique described in section 3.3?
- In Fig 3, are the authors modeling single-trials or condition averages (ie PSTHs)? This should be addressed. It looks like they are predicting condition averages since the "true" neural activity in 3E takes on continuous values (rather than indicating spike times or binned spike counts).
- Why does SRNN perform worse than SLDS and rSLDS in Figure 5CD?

**Limitations:**

The authors address several limitations, including their need to manually set the number of discrete states, their need for good parameter initializations, and the heavy computational requirements for fitting their models.

---

> ### Author Rebuttal · Authors · 2024-08-07
>
> We thank you deeply for your time and attention in reading our paper, and for your valuable comments. Below is our response to specific weaknesses and questions.
>
> **Weakness 1**: *Comparison between SNLDS and mrSDS*
>
> Thank you for raising this weakness. As also mentioned in the global rebuttal, we have now implemented these two models, the SNLDS and mrSDS. We show their performance on the Reaching Dataset in the attached PDF. To summarize, while recovering behaviorally-relevant states, our model SRNN outperforms SNLDS and has comparable performance with mrSDS, whereas in reconstructing neural activity, SRNN outperforms mrSDS and has comparable performance with SNLDS. **Therefore, SRNNs have overall better performance on recovering behaviorally-relevant states and reconstructing neural activity**. We would like to point out that mrSDS was published recently by ICLR 2024 in May, two weeks before our original submission, and to date, we have not found any code released with the paper. However, we implemented the code based on our reading and understanding of the paper to the best extent possible.
>
> **Weakness 2**: *Determining the appropriate number of discrete states. One approach to this might be co-smoothing*
>
> We thank you for your comments and for the excellent suggestion of ‘co-smoothing’. We have now implemented co-smoothing and find that SRNNs are successful at latent state modeling using this metric, and that this method is helpful for determining the appropriate number of discrete states. In addition, we now provide a comprehensive set of metrics to determine this important hyperparameter, including **co-smoothing** in the **global rebuttal** as well as in the official comment below. To summarize, we use appropriate selection metrics for the number of discrete states: (1) convergence / reuse of discrete states, (2) reconstruction performance and co-smoothing method, and (3) variability across conditions or trials. We show comprehensive results for these metrics on two experimental datasets where we recover the appropriate number of behaviorally relevant states using any of these three metrics.
>
> **Detail Weakness 1**: *The mathematical details and notation are often unclear. For example, eq 2: Shouldn't this instead be a categorical distribution or similar? Relatedly, f is also used in eq 8*
>
> Yes, you are right. It should be a softmax activation function and categorical distribution, which is what we implemented in the original paper. We apologize for the typo and we have revised both equation 2 and equation 8 in the paper.
>
> **Detail Weakness 2**: *The authors should more clearly describe the cross-validation techniques used for each dataset*
>
> Thank you for this comment and we apologize for the confusion. We are training the model on N-1 conditions/sessions/subjects and testing it on the 1 held-out condition/session/subject. In this process, we train N different models, one for each one of conditions/sessions/subjects on hold, and show the results for each model as one dot in Figures 3-5. We had detailed for each experimental dataset whether we are considering conditions or sessions or subjects to be held-out data, but have now moved it to the main text in the interest of clarity. Specifically, we hold out data from one condition at a time in the reaching dataset, where there are 18 different curved reaching conditions. We hold out data from one session at a time in the decision-making dataset, where there are 8 total sessions, and we hold out data from one subject at a time in the lever pull dataset, where there are 6 total subjects.
>
> **Q1)**: *Are the predictions in Fig 2 cross validated?*
>
> We thank you for your question and apologize for the lack of clarity. Yes, all models in this paper are cross-validated using held-out test sets. Specifically for Figure 2, we have one Lorenz attractor as the underlying dynamical system. We train SRNNs on 15 trials (representing different initial conditions), and test on 1 held-out trial. We have now clearly included this information in the paper.
>
> **Q2)**: *In Fig 3, are the authors modeling single-trials or condition averages (ie PSTHs)?*
>
> We are modeling condition averages in the paper. In addition to the analyses in the paper, we have now also trained SRNNs on simulated single trial data to assess if our method works on this kind of data. Specifically, we implemented the thinning algorithm using the time-rescaling theorem to simulate single trial spiking data based on the condition averaged firing rate [Brown et al., 2002]. We simulated 200 single trials from 5 condition averages and computed the binned spike counts, which we used to train the SRNN models. We found that the SRNNs successfully represent this data with P=16 and K=5; we show the results of training on simulated single trials in the attached PDF (Figure 6).
>
> **Q3)**: *Why does SRNN perform worse than SLDS and rSLDS in Fig 5CD?*
>
> In Figure 5C, the SRNNs indeed have worse reconstruction performance than SLDS and rSLDS. As we mentioned in our paper, ‘Indeed, all three methods have acceptable reconstruction.’: we didn’t find any significant difference by visualizing the reconstruction (Figure 5E of original paper). A potential reason causing the worse performance may be the small size of this dataset. As our response to Review 8VtM, a failure mode of SRNNs is when the model is trained in the low data regime, e.g., single conditions (see attached PDF). However, SRNNs have better recovery of behaviorally-relevant states. This may explain the finding that SRNNs have better prediction performance in 0.67, 1, and 1.33 seconds ahead in Figure 5D of the original paper; as the prediction window increases, the prediction performance may depend both on the recovered dynamics and the recovery of correct behavioral states. We believe that prediction of future activity provides a better test of a dynamical model, and here, SRNNs have overall better performance in prediction of neural activity.

---

> > ### Comment · Reviewer_hFy4 · 2024-08-12
> >
> > My primary concern remains only partially addressed: The contribution beyond other switching nonlinear dynamical systems models is not clear. I appreciate the attempts to implement and compare against SNLDS and mrSDS. And I appreciate that mrSDS was very recently accepted (though the paper has been on arxiv for many months before the ICLR acceptance) and that code may not have been publicly available. However, I expect a compelling description of the novelty of your approach relative to these existing approaches in the literature. What is the innovation beyond those approaches? And can you demonstrate that those specific innovations underlie the stated empirical performance benefits over SNLDS and mrSDS?

---

> ### Author Response · Authors · 2024-08-07
>
> **Details of Determining relevant hyperparameters (Weakness 2)**
>
> We now add a discussion for determining relevant hyperparameters, **such as the number of discrete and continuous latent states**. Importantly, we include a comprehensive hyperparameter sweep for the number of discrete states for both the reaching and decision-making datasets. We detail out the salient results for the reaching dataset, where there are 5 behavioral states in the task (decision-making results below):
>
> (1) **Convergence to lower number of discrete states**: We tested our model by increasing the number of hidden states K while keeping the number of continuous latent states P constant. We found that **61%** of SRNNs with a higher number of discrete hidden states (e.g., K=6) finally converge to the optimal number of discrete hidden states, i.e., K=5.
>
> (2) **Reuse of discrete states**: We also test our model by decreasing the number of hidden states. We found that **94%** of SRNNs with a lower number of discrete hidden states (e.g., K=4) had at least one hidden state reused after other states: in other words, SRNNs are not able to perform well with 4 unique discrete hidden states without reusing one of them.
>
> (3) **Reconstruction performance plateau**: While keeping other hyperparameters constant, the reconstruction accuracy plateaus at the same number of discrete states as in the behavior, thus we can use the minimum number of discrete states as it takes for the model to perform well. We have included a figure detailing this in the attached PDF (Figure 2). Moreover, we also implemented a ‘co-smoothing’ method, we show the results in the attached PDF (Figure 3), where we found that K=5 also does well in reconstructing the data with a ‘co-smoothing’ neuron drop-out analysis.
>
> (4) **Variability across conditions**: In stereotyped tasks or experiments, such as reaching, there may not be a significant amount of variability in the timing of behavior across conditions, and this variability can thus be used as a metric for determining the number of discrete states. **Here, we found that SRNNs with K=5 have much lower variability on recovered behaviorally-relevant states than K=4 and K=6 (i.e., 0.098 for K=5, 0.384 for K=4, and 0.282 for K=6).**
>
> Furthermore, we did the same analysis for the decision-making data (Figure 4 in the attached PDF) and we found the same situation as in the reaching dataset. **Higher K also converges to a smaller number of discrete states, and K=5 has the smallest variability across different pseudo-sessions (i.e., 0.106 for K=5, 0.287 for K=4, and 0.290 for K=6).**
>
> These results demonstrate appropriate selection metrics for the number of discrete states: (1) convergence / reuse of discrete states, (2) reconstruction performance, and (3) variability across conditions or trials. We show comprehensive results for these metrics on two experimental datasets where we recover the appropriate number of behaviorally relevant states using any of these three metrics. We have now provided this in the paper, while adding a discussion for the general case.
>
> Additionally, we show a comparison between values for another important hyperparameter P in the original submission.

---

> ### Author Response · Authors · 2024-08-12
>
> We thank you for your response. We include a detailed feature comparison between SNLDS, mrSDS, and SRNNs in the following table. Specifically, SRNNs differ fundamentally from SNLDS and mrSDS in their **generative model, inference network**, and **initialization**, as detailed below. Moreover, we examine whether these models, trained directly on neural data, can reconstruct _**behaviorally-relevant discrete states**_: a question that has not been investigated in either of the previous studies. We found that, without fail, **SRNNs have better performance** in this goal than competing methods, including SNLDS and mrSDS.
>
> (1) **Generative Model**: SRNNs utilize RNN as a generative model, while SNLDS and mrSDS employ a MLP with nonlinear activation functions. RNNs may offer greater interpretability and accuracy compared to feedforward models, and are very relevant to the neuroscience community for insights on computational mechanisms [1][2].
>
> (2) **Inference Network**: mrSDS uses a non-causal transformer encoder as an inference network and SNLDS uses a non-causal combination of RNNs as an inference network. SRNNs are able to **predict** neural activity many time-steps ahead since we have implemented both **causal and non-causal** RNNs as possible inference networks. This important capability brings us one step closer to real-time and closed-loop neuroscience applications.
>
> (3) **Initialization**: SRNN has a different initialization method compared to SNLDS and mrSDS (*unknown*). In training SRNNs, we rely on HMMs, whereas SNLDS uses uniform entropy regularization. We would like to clarify that the initialization of mrSDS is unknown, therefore, we used both initialization approaches when we implemented mrSDS and report the better performance for mrSDS.
>
> **In summary, our primary contributions are**: (1) characterizing and interpreting low-dimensional switching nonlinear neural dynamics using SRNNs, (2) enabling causal prediction of neural activity, and (3) demonstrating that SRNNs outperform SNLDS and mrSDS in _**recovering behaviorally-relevant states and reconstructing and predicting neural activity**_, making them more reliable for interpreting behaviorally-relevant neural dynamics.
>
> We thank you again for your questions. We hope this comparison is compelling and we look forward to your response for further discussion if needed.
>
> | Models (on Reaching Data) | Generative Model | Inference Network  |Initialization | Behavioral States Recovery (**Error**)  |Reconstruction (**MSE**)| Multi-regional setting | Neural Prediction | Code |
> | :----:       |    :----:   |        :----: |    :----:   |        :----: |    :----:   |        :----: |    :----:   |     :----: |
> | SNLDS      | MLP       | NNs   |Entropy regularization| 0.46 ($\pm$0.125)      | 0.00404 ($\pm$0.00170)       | No  |Not implemented (non-causal inference network)|Tensorflow|
> | mrSDS   | MLP        | Transformer      |*Unknown*| 0.32 ($\pm$0.139)      | 0.00806 ($\pm$0.00273)       | **Yes**   |Not implemented (non-causal inference network)|*Unknown*|
> | SRNN (Ours)   | RNN       | NNs    |HMM| **0.27 ($\pm$0.093)**      | **0.00230 ($\pm$0.00041)**      | **Yes**   | **Yes** (causal inference network)|Pytorch|
>
> [1] Durstewitz, Daniel, Georgia Koppe, and Max Ingo Thurm. "Reconstructing computational system dynamics from neural data with recurrent neural networks." Nature Reviews Neuroscience 24.11 (2023): 693-710.
>
> [2] Barak, Omri. "Recurrent neural networks as versatile tools of neuroscience research." Current opinion in neurobiology 46 (2017): 1-6.

---

> > ### Comment · Reviewer_hFy4 · 2024-08-12
> >
> > I have read the authors' responses as well as the comments from the other reviewers. I stand by my rating of 3 due to the limited originality and innovation of the contribution. I do not see the impact of this work meeting the high standard of NeurIPS. While the authors did describe the differences between their approach and previous related approaches, the novelty appears quite limited, and the authors have not demonstrated which of these differences (if any) lead to improved performance.

---

### Official Review · Reviewer_8VtM · 2024-07-13

**Soundness:** 3
**Presentation:** 3
**Contribution:** 2
**Rating:** 6
**Confidence:** 3

**Summary:**

The paper proposes switching recurrent neural networks (SRNN), which allow the RNN weights to switch across time. The RNN weights switch based on a latent Markovian process of discrete states. The authors apply SRNN to a simulated dataset following the Lorenz attractor and three real-world neural recordings.

**Strengths:**

- Clarity: The authors clearly explain the problem, related work, and methodology with well-written equations and easy-to-understand figures.

- Extensive use of datasets: The paper applies SRNN to numerous real-world neural datasets, illustrating the effectiveness of SRNN in accurately segmenting different datasets in an unsupervised fashion.

**Weaknesses:**

- Lack of comparison with other methods:
The paper compares SRNN to (r)SLDS models. However, there exist many other models for unsupervised segmentation. For example, ARHMMs and their extensions are simple yet powerful and interpretable models for segmentation [1, 2]. The authors should cite and consider comparisons with multiple model classes.
In addition, the paper notes in line 103 that SRNNs have the most comparable structure to SNLDS, but the authors do not make comparisons. The authors should also cite and compare with [3], which has switching nonlinear dynamics.

[1] Wiltschko, A. B., Johnson, M. J., Iurilli, G., Peterson, R. E., Katon, J. M., Pashkovski, S. L., ... & Datta, S. R. (2015). Mapping sub-second structure in mouse behavior. Neuron, 88(6), 1121-1135.

[2] Lee, H. D., Warrington, A., Glaser, J., & Linderman, S. (2023). Switching autoregressive low-rank tensor models. Advances in Neural Information Processing Systems, 36, 57976-58010.

[3] Karniol-Tambour, O., Zoltowski, D. M., Diamanti, E. M., Pinto, L., Tank, D. W., Brody, C. D., & Pillow, J. W. (2022). Modeling communication and switching nonlinear dynamics in multi-region neural activity. bioRxiv, 2022-09.

- Experiments:
The simulated experiment with the Lorenz attractor shows that SRNN does well when it has access to noiseless observations with known state dimensions. In order to have a more convincing simulated experiment, the authors could consider the following. First project the Lorenz attractor to a higher dimensional space and add additive Gaussian noise. Then fit SRNN (and other compared models) to the dataset to see if it can recover the Lorenz attractor and true latent state dimension (using some metric on held-out data). Another simulated experiment could be done with a dataset that simulates the NASCAR track [1,2].

[1] Linderman, S. W., Miller, A. C., Adams, R. P., Blei, D. M., Paninski, L., & Johnson, M. J. (2016). Recurrent switching linear dynamical systems. arXiv preprint arXiv:1610.08466.

[2] Lee, H. D., Warrington, A., Glaser, J., & Linderman, S. (2023). Switching autoregressive low-rank tensor models. Advances in Neural Information Processing Systems, 36, 57976-58010.

**Questions:**

- What are some failure modes of the model? Does extra flexibility mean that SRNNs need more data than simpler models such as ARHMMs or SLDSs? I'm curious how the SRNNs would do in a low-data regime (e.g., sample a small amount of dataset from an SLDS).

- Have you tried fitting the model to datasets other than neural data? Based on how well SRNNs do in segmenting the neural datasets, I'm curious how SRNNs would do on other types of datasets, such as mouse behavioral dataset [1].

[1] Wiltschko, A. B., Johnson, M. J., Iurilli, G., Peterson, R. E., Katon, J. M., Pashkovski, S. L., ... & Datta, S. R. (2015). Mapping sub-second structure in mouse behavior. Neuron, 88(6), 1121-1135.

- How are the hyperparameters selected? Based on how long it takes to fit each SRNN to the datasets, I wonder if it is feasible to sweep over the hyperparameter space.

**Limitations:**

As the authors noted, some limitations of the model are that the model needs good initialization and that the model takes considerably more amount of time to train than simpler models such as SLDSs.

---

> ### Author Rebuttal · Authors · 2024-08-07
>
> We thank you very much for giving us a positive rating, and for you very helpful comments. Below we respond to specific weaknesses pointed out.
>
> **Weakness 1**: *Lack of comparison with other methods, for example, ARHMMs and their extensions, as well as SNLDS and mrSDS*
>
> We have now included the two suggested baseline models, mrSDS and SNLDS, in the PDF attached above. To summarize, as above, while recovering behaviorally-relevant states, our model SRNN outperforms SNLDS and has comparable performance with mrSDS, whereas in reconstructing neural activity, SRNN outperforms mrSDS and has comparable performance with SNLDS . **Therefore, SRNNs have overall better performance on recovering behaviorally-relevant states and reconstructing neural activity**. Additionally, we would like to point out that we used the SSM package to install SLDS and rSLDS, where all models are initialized with ARHMM, and thus expect that the log likelihoods are comparable or higher in these baseline models, already included in the initial submission.
>
> **Weakness 2**: *Experiments: the simulated experiment with the Lorenz attractor shows that SRNN does well when it has access to noiseless observations with known state dimensions. In order to have a more convincing simulated experiment, the authors could consider the following. First project the Lorenz attractor to a higher dimensional space and add additive Gaussian noise. Then fit SRNN (and other compared models) to the dataset to see if it can recover the Lorenz attractor and true latent state dimension (using some metric on held-out data). Another simulated experiment could be done with a dataset that simulates the NASCAR track*
>
> We thank the reviewer for suggesting the noisy high dimensional Lorenz Attractor and the common simulated data NASCAR. We now include results on testing SRNNs using the suggested setup for the noisy high dimensional Lorenz Attractor and NASCAR in the PDF attached above (Figure 9 and Figure 10). To simulate the Lorenz Attractor, we project the 3-dimensional Lorenz into an 8-dimensional space and add additive Gaussian noise. We found that SRNNs with P=3 can successfully recover the butterfly structure of a Lorenz Attractor using the noisy high dimensional data. Moreover, we found that our SRNNs can successfully recover the ground truth of the trajectories and states of the NASCAR dataset.
>
> **Q1)** *What are some failure modes of the model? Does extra flexibility mean that SRNNs need more data than simpler models such as ARHMMs or SLDSs? I'm curious how the SRNNs would do in a low-data regime*
>
> We thank you for your question. One failure mode may be when the training data is limited. To identify this, we trained our model on a single condition in the reaching dataset instead of 26 conditions; we found that the test reconstruction accuracy is lower than for SLDS and rSLDS (see attached PDF, Fig 5). Another failure mode may be the commonly-encountered problem of overfitting: we found that SRNNs may overfit to the training data if we increase the number of parameters (number of states K and number of latents P) to be large. Lastly, there may be a small problem of identifiability as pointed out by Reviewer Kpfw, i.e., if we increase the number of latents to be very large, the neural data can be successfully modeled by a single RNN, but of course this fails to recover interpretable switches in neural dynamics. We identify these tradeoffs and provide a discussion in the rebuttal for Reviewer Kpfw.
>
> **Q2)** *Have you tried fitting the model to datasets other than neural data? Based on how well SRNNs do in segmenting the neural datasets, I'm curious how SRNNs would do on other types of datasets, such as mouse behavioral dataset*
>
> We agree with the reviewer that behavioral data may provide the interpretability that we desire while modeling. We tested SRNNs on poses tracked using DeepLabCut, specifically the nose and paw positions, as recorded by a behavioral camera in the head-fixed decision-making task (see Fig. 4C in Rebuttal PDF). Interestingly, the states recovered were not as accurate as when using the neural data (see Fig. 4 of original submission). In this task, a visual stimulus was presented either to the left or the right, not visible in the camera. Specifically, the model trained on behavioral pose data failed to identify a separate state for ‘stimulus presentation’, presumably since the nose and paw location was not noticeably different here than during the delay period. This exercise points towards the possibility of augmenting behavioral observations with neural activity in future work, to better inform switching states.
>
> **Q3)** *How are the hyperparameters selected? Based on how long it takes to fit each SRNN to the datasets, I wonder if it is feasible to sweep over the hyperparameter space*
>
> One important parameter is the number of discrete states K, and we show a comparison between different numbers of hidden states in the PDF attached and include a thorough discussion in the **global rebuttal**. Another very important hyperparameter is the number of hidden states P of SRNN, we show a comparison between different numbers of hidden states in the original submission. We included the discussion about determining K and P in the global rebuttal as well as in the official comment below. For other hyperparameters, such as the number of hidden units in our inference network and learning rate, we used cross validation and kept the parameters with best performance, but we found that these other parameters did not affect the results very much. It is possible to sweep over the hyperparameter space, however, this may require training in parallel with large resources, for example, in the reaching dataset, training one model with P=16 and K=5 takes 9 hours 32 mins on a NVIDIA A100 GPU, and training models with (P=2, K=5), (P=4, K=5), and (P=8, K=5) takes 8 hours 01 min, 8 hours 30 mins, and 8 hours 54 mins respectively.

---

> ### Author Response · Authors · 2024-08-07
>
> **Details of Determining relevant hyperparameters (Q3)**
>
> We now add a discussion for determining relevant hyperparameters, **such as the number of discrete and continuous latent states**. Importantly, we include a comprehensive hyperparameter sweep for the number of discrete states for both the reaching and decision-making datasets. We detail out the salient results for the reaching dataset, where there are 5 behavioral states in the task (decision-making results below):
>
> (1) **Convergence to lower number of discrete states**: We tested our model by increasing the number of hidden states K while keeping the number of continuous latent states P constant. We found that **61%** of SRNNs with a higher number of discrete hidden states (e.g., K=6) finally converge to the optimal number of discrete hidden states, i.e., K=5.
>
> (2) **Reuse of discrete states**: We also test our model by decreasing the number of hidden states. We found that **94%** of SRNNs with a lower number of discrete hidden states (e.g., K=4) had at least one hidden state reused after other states: in other words, SRNNs are not able to perform well with 4 unique discrete hidden states without reusing one of them.
>
> (3) **Reconstruction performance plateau**: While keeping other hyperparameters constant, the reconstruction accuracy plateaus at the same number of discrete states as in the behavior, thus we can use the minimum number of discrete states as it takes for the model to perform well. We have included a figure detailing this in the attached PDF (Figure 2). Moreover, we also implemented a ‘co-smoothing’ method as suggested by Reviewer hFy4 [Yu et al., 2009 and Karniol-Tambour et al., ICLR 2024], we show the results in the attached PDF (Figure 3), where we found that K=5 also does well in reconstructing the data with a ‘co-smoothing’ neuron drop-out analysis.
>
> (4) **Variability across conditions**: In stereotyped tasks or experiments, such as reaching, there may not be a significant amount of variability in the timing of behavior across conditions, and this variability can thus be used as a metric for determining the number of discrete states. **Here, we found that SRNNs with K=5 have much lower variability on recovered behaviorally-relevant states than K=4 and K=6 (i.e., 0.098 for K=5, 0.384 for K=4, and 0.282 for K=6).**
>
> Furthermore, we did the same analysis for the decision-making data (Figure 4 in the attached PDF) and we found the same situation as in the reaching dataset. **Higher K also converges to a smaller number of discrete states, and K=5 has the smallest variability across different pseudo-sessions (i.e., 0.106 for K=5, 0.287 for K=4, and 0.290 for K=6).**
>
> These results demonstrate appropriate selection metrics for the number of discrete states: (1) convergence / reuse of discrete states, (2) reconstruction performance, and (3) variability across conditions or trials. We show comprehensive results for these metrics on two experimental datasets where we recover the appropriate number of behaviorally relevant states using any of these three metrics. We have now provided this in the paper, while adding a discussion for the general case.
>
> Additionally, we show a comparison between values for another important hyperparameter P in the original submission.

---

> > ### Comment · Reviewer_8VtM · 2024-08-09
> >
> > I would like to thank the authors for their response to my comments and clarifications. I would like to raise my score from 5 (Borderline accept) to 6 (Weak accept).
> >
> > However, I still believe that there needs to be comparisons with simpler models such as ARHMMs and their extensions [1, 2]. In particular, I do not agree with "...SLDS and rSLDS, where all models are initialized with ARHMM, and thus expect that the log-likelihoods are comparable or higher in these baseline models...". SLDS and rSLDS rely on approximate inference which may frequently lead to bad optima. In contrast, ARHMMs sometimes tend to perform better, thanks to their exact M-step update.
> >
> > [1] Wiltschko, A. B., Johnson, M. J., Iurilli, G., Peterson, R. E., Katon, J. M., Pashkovski, S. L., ... & Datta, S. R. (2015). Mapping sub-second structure in mouse behavior. Neuron, 88(6), 1121-1135.
> >
> > [2] Lee, H. D., Warrington, A., Glaser, J., & Linderman, S. (2023). Switching autoregressive low-rank tensor models. Advances in Neural Information Processing Systems, 36, 57976-58010.
> >
> > If accepted, I highly suggest including these additional comparisons with ARHMMs and their variants.

---

> ### Author Response · Authors · 2024-08-10
>
> >*I would like to thank the authors for their response to my comments and clarifications. I would like to raise my score from 5 (Borderline accept) to 6 (Weak accept).*
>
> >*However, I still believe that there needs to be comparisons with simpler models such as ARHMMs and their extensions [1, 2]. In particular, I do not agree with "...SLDS and rSLDS, where all models are initialized with ARHMM, and thus expect that the log-likelihoods are comparable or higher in these baseline models...". SLDS and rSLDS rely on approximate inference which may frequently lead to bad optima. In contrast, ARHMMs sometimes tend to perform better, thanks to their exact M-step update.*
>
> >*[1] Wiltschko, A. B., Johnson, M. J., Iurilli, G., Peterson, R. E., Katon, J. M., Pashkovski, S. L., ... & Datta, S. R. (2015). Mapping sub-second structure in mouse behavior. Neuron, 88(6), 1121-1135. *
> >*[2] Lee, H. D., Warrington, A., Glaser, J., & Linderman, S. (2023). Switching autoregressive low-rank tensor models. Advances in Neural Information Processing Systems, 36, 57976-58010.*
>
> >*If accepted, I highly suggest including these additional comparisons with ARHMMs and their variants.*
>
> We thank you for your response and raising the score from 5 to 6. We also thank you for pointing out that “ARHMMs sometimes tend to perform better, thanks to their exact M-step update.” Yes, you are right, we applied ARHMM on the reaching data and found that ARHMM outperform SLDS and rSLDS in the reconstruction of neural activity. Therefore, we included a comparison between SRNNs vs ARHMMs on reaching data are as following:
>
> |    | SRNN(P=16, K=5) | ARHMMs (R=180, K=5)    |
> | :---        |    :----:   |          ---: |
> | Reconstruction (**MSE**)      |   **0.00230($\pm$0.00041)**   |          0.00232($\pm$0.00085) |
> | Behavioral States Recovery (**Error**)      | **0.27($\pm$0.093)**       |  0.78($\pm$0.22)   |
> | Training on Single Condition (**MSE**)   | **0.01117($\pm$0.00099)**        | 0.01216($\pm$0.00104)     |
>
> We found that SRNNs and ARHMMs exhibit very similar reconstruction accuracy, both outperform SLDS and rSLDS. However, SRNNs show a slight advantage in terms of the mean and standard deviation of the mean squared error (MSE) between neural activity and the reconstructions. Despite their performance in reconstruction, ARHMMs are not able to recover behaviorally relevant states, and the states identified by ARHMMs lack interpretability.
>
> We agree that the extensions of ARHMMs, such as Switching autoregressive low-rank tensor models, are interesting to compare. We will include a comprehensive comparison between all these models mentioned above in the final version, if accepted.

---

### Author Rebuttal · Authors · 2024-08-07

Dear Reviewers,

We would like to thank you for providing constructive feedback that helped us improve the paper. As a reminder, in this submission, we propose ‘Switching Recurrent Neural Networks’ (SRNNs) for discovery of switching neural dynamics that leads to behaviorally-relevant discrete states, and leads to more accurate reconstruction and prediction of neural data than baseline methods. Here, we detail responses to suggestions made by multiple reviewers, and address individual reviews below. If any questions are unanswered or our responses are unclear, we would appreciate the chance to engage further with you. Additionally, **please find a PDF with helper figures attached**. These are referenced and described in our responses.

### **1. Additional baseline models**
In addition to the **existing SLDS, rSLDS, and LFADS**, we implemented and have updated our paper with two more baseline models that reviewers mentioned, the **SNLDS** [Dong et al., ICML 2020] and **mrSDS** [Karniol-Tambour et al., ICLR 2024]. We show their performance on the Reaching Dataset in the attached PDF (Figure 1). To summarize, while recovering behaviorally-relevant states, our model SRNN outperforms SNLDS and has comparable performance with mrSDS, whereas in reconstructing neural activity, SRNN outperforms mrSDS and has comparable performance with SNLDS. **Therefore, SRNNs have overall better performance on recovering behaviorally-relevant states and reconstructing neural activity**. We would like to point out that mrSDS was published recently by ICLR 2024 in May, two weeks before our original submission, and to date, *we have not found any code released with the paper*. However, we implemented the code based on our reading and understanding of the paper to the best extent possible.

### **2. Determining relevant hyperparameters**
We now add a discussion for determining relevant hyperparameters, **such as the number of discrete and continuous latent states**. Importantly, we include a comprehensive hyperparameter sweep for the number of discrete states for both the reaching and decision-making datasets. We detail the salient results for the reaching dataset, where there are 5 behavioral states in the task (decision-making results below):

(1) **Convergence to lower number of discrete states**: We tested our model by increasing the number of hidden states K while keeping the number of continuous latent states P constant. We found that **61%** of SRNNs with a higher number of discrete hidden states (e.g., K=6) finally converge to the optimal number of discrete hidden states, i.e., K=5.

(2) **Reuse of discrete states**: We also test our model by decreasing the number of hidden states. We found that **94%** of SRNNs with a lower number of discrete hidden states (e.g., K=4) had at least one hidden state reused after other states: in other words, SRNNs are not able to perform well with 4 unique discrete hidden states without reusing one of them.

(3) **Reconstruction performance plateau**: While keeping other hyperparameters constant, the reconstruction accuracy plateaus at the same number of discrete states as in the behavior, thus we can use the minimum number of discrete states as it takes for the model to perform well. We have included a figure detailing this in the attached PDF (Fig 2). Moreover, we also implemented a ‘co-smoothing’ method as suggested by Reviewer hFy4 [Yu et al., 2009 and Karniol-Tambour et al., ICLR 2024], we show the results in the attached PDF (Fig 3), where we found that K=5 also does well in reconstructing the data with a ‘co-smoothing’ neuron drop-out analysis.

(4) **Variability across conditions**: In stereotyped tasks or experiments, such as reaching, there may not be a significant amount of variability in the timing of behavior across conditions, and this variability can thus be used as a metric for determining the number of discrete states. **Here, we found that SRNNs with K=5 have much lower variability on recovered behaviorally-relevant states than K=4 and K=6 (i.e., 0.098 for K=5, 0.384 for K=4, and 0.282 for K=6)**.

Furthermore, we did the same analysis for the decision-making data (Fig 4 in the attached PDF) and we found the same situation as in the reaching dataset. **Higher K also converges to a smaller number of discrete states, and K=5 has the smallest variability across different pseudo-sessions (i.e., 0.106 for K=5, 0.287 for K=4, and 0.29 for K=6)**.

These results demonstrate appropriate selection metrics for the number of discrete states: (1) convergence / reuse of discrete states, (2) reconstruction performance, and (3) variability across conditions or trials. We show comprehensive results for these metrics on two experimental datasets where we recover the appropriate number of behaviorally relevant states using any of these three metrics. We have now provided this in the paper, while adding a discussion for the general case.

Finally, we show a comparison between values for another important hyperparameter P in the original submission.

### **3. Additional analyses**

In response to **individual reviewers**, we also implemented the following which have made our paper much stronger.

(1) SRNN trained on single reaching conditions to explore failure modes of our model

(2) SRNN trained on simulated single trials of binned spike counts

(3) Comparison between SRNN and a standard RNN with a higher dimensional continuous latent state to assess identifiability of the models

(4) Flow fields with a different random seed to assess stability of recovered dynamics

(5) SRNN on two more simulated datasets: high dimensional noisy Lorenz Attractor and NASCAR

### **4. Typos in Equations and Editing of Paper**
We appreciate all comments and suggestions on pointing out the typos and errors of the paper. We apologize for the inconvenience during the review process.

We thank all reviewers for the suggestions and comments and provide a detailed rebuttal to each reviewer below.

---

### Author Response · Authors · 2024-08-10
**A Potential Glitch**

Dear Reviewers,

Due to a potential glitch, the reviewer comments might not be visible to all reviewers. Therefore, we have included the full reviewer comments in our response. Thank you!

---

### Decision · Program_Chairs · 2024-09-25

**Decision:**

Accept (poster)

**Comment:**

This paper describes a novel approach to modeling neural dynamics using switching RNNs.  The paper generated considerable interest and discussion among the reviewers. Ultimately, I am persuaded by the comments of the more positive reviewers, and feel that it should be accepted to the meeting. Please thoroughly revise the manuscript to address all reviewer comments and discussion points.